# Polygodial, a Sesquiterpene Dialdehyde, Activates Apoptotic Signaling in Castration-Resistant Prostate Cancer Cell Lines by Inducing Oxidative Stress

**DOI:** 10.3390/cancers14215260

**Published:** 2022-10-26

**Authors:** Reshmii Venkatesan, Mohamed Ali Hussein, Leah Moses, Jennifer S. Liu, Salman R. Khetani, Alexander Kornienko, Gnanasekar Munirathinam

**Affiliations:** 1Department of Biomedical Sciences, College of Medicine, University of Illinois, 1601 Parkview Avenue, Rockford, IL 61107, USA; 2Department of Biomedical Engineering, University of Illinois at Chicago, Chicago, IL 60607, USA; 3Department of Chemistry and Biochemistry, Texas State University, San Marcos, TX 78666, USA

**Keywords:** prostate cancer, castration-resistant prostate cancer, polygodial, taxane-resistant CRPC, anoikis, natural products

## Abstract

**Simple Summary:**

Prostate cancer (PCa) is the second leading cause of cancer death in men in the United States. The emergence of resistance to androgen deprivation therapy results in castration-resistant prostate cancer (CRPC) development. Taxanes are diterpene compounds approved to treat hormonal-resistant PCa. CRPC patients treated with taxanes show poor outcomes. Polygodial (PG) is a natural sesquiterpene compound isolated from water pepper (*Persicaria hydropiper*), Dorrigo pepper (*Tasmannia stipitata*), and mountain pepper (*Tasmannia lanceolata*), which has shown to exhibit anticancer properties. PG robustly inhibits the viability, colony formation, and migration of taxane-resistant CRPC (PC3-TXR and DU145-TXR) cell lines. Additionally, our data show that PG promotes anoikis and induces cell cycle arrest at the G0 phase in PCa cells. Our results reveal that PG induces oxidative stress and activates apoptosis in drug-resistant CRPC cell lines. Altogether, our data suggest that the anticancer activity of PG is via the induction of apoptosis in CRPC cells.

**Abstract:**

Prostate cancer (PCa) is the second leading cause of cancer death among men in the United States. Surgery, radiation therapy, chemotherapy, and androgen deprivation therapy are currently the standard treatment options for PCa. These have poor outcomes and result in the development of castration-resistant prostate cancer (CRPC), which is the foremost underlying cause of mortality associated with PCa. Taxanes, diterpene compounds approved to treat hormonal refractory PCa, show poor outcomes in CRPC. Polygodial (PG) is a natural sesquiterpene isolated from water pepper (*Persicaria hydropiper*), Dorrigo pepper (*Tasmannia stipitata*), and mountain pepper (*Tasmannia lanceolata*). Previous reports show that PG has an anticancer effect. Our results show that PG robustly inhibits the cell viability, colony formation, and migration of taxane-resistant CRPC cell lines and induces cell cycle arrest at the G0 phase. A toxicity investigation shows that PG is not toxic to primary human hepatocytes, 3T3-J2 fibroblast co-cultures, and non-cancerous BPH-1 cells, implicating that PG is innocuous to healthy cells. In addition, PG induces oxidative stress and activates apoptosis in drug-resistant PCa cell lines. Our mechanistic evaluation by a proteome profiler–human apoptotic array in PC3-TXR cells shows that PG induces upregulation of cytochrome c and caspase-3 and downregulation of antiapoptotic markers. Western blot analysis reveals that PG activates apoptotic and DNA damage markers in PCa cells. Our results suggest that PG exhibits its anticancer effect by promoting reactive oxygen species generation and induction of apoptosis in CRPC cells.

## 1. Introduction

Prostate cancer (PCa) is the most common cancer among men in the United States of America (USA), with an estimated 268,490 new cases in 2022, and it occupies the second place in causes of high mortality among men in the USA, after lung cancer [1,2]. One in eight men will have PCa during their lifetime, and the five-year relative survival rate for PCa is 96.8% [2]. The global burden of PCa is vastly increased worldwide, with PCa ranked fourth among the top five most common cancers, with 7.3% of all new cancer cases being PCa [3]. The incidence of PCa increases in older men aged above 50 years, men of African descent, men with a family history of the disease, and men with genetic risk factors [4]. The early stages of PCa are indolent and mostly asymptomatic; by the time it becomes symptomatic, it is aggressive and metastasizes to other parts of the body [4]. The heterogeneity of PCa is due to the disease’s multifaceted nature, which mainly originates from the spatial, morphological, genetic, and molecular differences of PCa. The genetic and molecular aspects particularly show discrepancies due to intra-patient, inter-patient, and intra-tumoral variations, rendering PCa a complex disease and making targeted genetic therapy impermissible [5,6].

The current armamentarium in treating PCa includes the usage of different strategies such as surgery (surgical castration), radiation therapy (external beam radiotherapy), androgen deprivation therapy (ADT) inducing castration using luteinizing hormone-releasing hormone agonist or antagonist (leuprorelin and goserelin), antiandrogen (abiraterone, enzalutamide, and bicalutamide) and nonsteroidal antiandrogen (such as apalutamide), chemotherapy including carboplatin, and taxanes (paclitaxel, docetaxel, and cabazitaxel), immunotherapy (sipuleucel-T and pembrolizumab), or a combination of these treatment options [7,8]. Surgery and radiation therapy are the most preferable options for localized PCa [9,10,11]. ADT is used as a primary systematic treatment for regional or advanced PCa, or as an adjuvant/neoadjuvant therapy in combination with radiotherapy in locally advanced PCa [8]. Taxanes, including paclitaxel, docetaxel, and cabazitaxel, approved to treat advanced and hormonal refractory PCa, show poor outcomes in CRPC [12,13]. Resistance to ADT results in the attainment of castration-resistant prostate cancer (CRPC), which is identified as progression of the disease with a concomitant upsurge in the serum levels of the prostate-specific antigen (PSA), despite using ADT therapy [14]. The CRPC spectrum ranges from a surge in the PSA serum level alone, to an increase in the PSA level and metastasis, giving rise to a more progressive form called metastatic CRPC [15]. Predominantly, resistance to ADT therapy emerges within two to three years of treatment [16]. The current options available for the treatment of metastatic CRPC are ADT in combination with luteinizing hormone-releasing hormone agonist-antagonist, along with taxanes [8]. However, taxanes show poor outcomes, resistance, and severe side effects [17,18]. Therefore, there is an urgent need for a better treatment approach for men with CRPC.

Over the years, natural products or their derivatives have shown chemopreventive and chemotherapeutic benefits [19,20,21]. The indefinite diversity of natural products renders diverse biological functions that may be advantageous over conventional chemotherapeutic agents [22,23,24]. Several natural products have been shown to regulate the oncogene and tumor suppressor genes in cancer [25], epigenetic mechanisms [26], and tumor microenvironment [27]. Polygodial (PG) is a sesquiterpene isolated from water pepper (*Persicaria hydropiper*), Dorrigo pepper (*Tasmannia stipitata*), and mountain pepper (*Tasmannia lanceolata*) [28,29]. PG exhibits antibacterial activity against Gram-negative bacteria, such as *Escherichia coli* and *Salmonella choleraesuis,* and Gram-positive bacteria, such as *Bacillus subtilis* and *Staphylococcus aureus* [29]. PG also exhibits antifungal, anti-inflammatory, and anticancer activity [29,30,31]. Several studies have reported that PG and its derivatives, including isopolygodial, 9-epipolygodial (DR-P27), 1-β-(p-coumaroyloxy)-polygodial, and 1-β-(*p*-methoxycinnamoyl)-polygodial, exhibit anticancer activity in different cancers [32,33,34,35]. Dasari et al. [30] reported that DR-P27 possesses superior antiproliferative efficacy compared to PG in drug resistant cancer cells. In contrast, De La Chapa et al. [32] showed that DR-P27 is equipotent to PG in oral squamous cell carcinoma (OSCC) [32]. Our previous study has demonstrated that the PG derivative DR-P27 exhibits anticancer activity against androgen-sensitive PCa by inducing apoptosis in vitro [28]. In this study, we have examined the anticancer efficacy and the underlying anticancer mechanism of PG against the taxane-resistant CRPC using an in vitro model. In addition, our study revealed for the first time that PG treatment induces anoikis in CRPC cell lines.

## 2. Materials and Methods

### 2.1. Cell Lines

PCa cell lines (PC3-TXR and DU145-TXR) were generously donated by Dr. Evan Keller (University of Michigan, Ann Arbor, Michigan, USA). These cells were established from PC3 and DU145 through exposure to paclitaxel at regular intervals, as described by Takeda et al. [36]. The BPH-1 cell line was obtained from Dr. Simon W. Hayward (University of Chicago, Northshore Research Institute, Pritzker School of Medicine, Chicago, USA). The BPH-1 cells were cultured in RPMI-1640 media (Gibco, Grand Island, NY, USA) supplemented with 10% (*v/v*) fetal bovine serum (FBS), 1% (*v/v*) antimycotic (Sigma Aldrich, St. Louis, MO, USA), and 0.6% (*v/v*) gentamycin (Fisher Scientific, Waltham, MA, USA) and grown at 37 °C with 5% CO_2_. Micropatterned co-cultures (MPCCs) of primary human hepatocytes and 3T3-J2 murine embryonic fibroblasts were created as described previously [37]. The cells were seeded into the micropatterned plates in a serum-free culture medium, consisting of 1X Dulbecco’s modified Eagle’s medium (DMEM, Corning Life Sciences, Corning, NY, USA) supplemented with 15 mM HEPES [4-(2-hydroxyethyl)-1-piperazineethane-sulfonic acid] buffer (Corning Life Sciences, Corning, NY, USA), 1% penicillin/streptomycin, 1% ITS+ (insulin, transferrin, selenous acid, linoleic acid, bovine serum albumin; Corning Life Sciences, Corning, New York, NY, USA), 7 ng mL-1 glucagon (Sigma-Aldrich, St. Louis, MO, USA), and 0.1 µM dexamethasone (Sigma-Aldrich, St. Louis, MO, USA). The hepatocytes preferentially attached to the circular collagen domains, leaving approximately 5000 primary human hepatocytes per 96 well plates. The 3T3-J2 murine embryonic fibroblasts were generously donated by Dr. Howard Green [38]. The cells were cultured in DMEM containing 10% bovine calf serum and 1% penicillin/streptomycin.

### 2.2. Chemicals and Antibodies

PG with ≥97% purity was obtained from Sigma-Aldrich (Sigma-Aldrich, St. Louis, MO, USA). The molecular weight of PG is 234.33 g/mol. The main stock of 10 mM was prepared by dissolving the drug in 0.1% dimethyl sulfoxide (DMSO), purchased from Fisher Scientific (Waltham, MA, USA). Further, different concentrations were prepared from the main stock and added to complete RPMI for treating cells. Necrostatin-1 (NEC), 3–4,5-dimethylthiazol-2-yl,2,5-diphenyl tetrazolium bromide (MTT), N-acetyl cysteine (NAC), propidium Iodide (PI), and 3-methyladenine (3MA) was obtained from Sigma-Aldrich, St. Louis, MO, USA. The pan caspase inhibitor (CI) Z-VAD-FMK was procured from APExBIO (Houston, TX, USA). The antibodies used were against PARP, pH2AX, Bcl-2, caspase-3, cleaved caspase-3, pro-caspase-3, XIAP, cIAP-2, and β-actin. HRP-conjugated anti-rabbit and HRP-conjugated anti-mouse antibodies were obtained from Cell Signaling Technology (Danvers, MA, USA).

### 2.3. Cell Viability Assay

To examine the effect of PG on the cell viability of PCa cell lines (PC3-TXR and DU145-TXR), cells were treated with various concentrations of PG. Cell viability was measured using MTT dye to determine the colorimetric dye reduction, according to van Meerloo et al. [39]. The PCa cells were seeded in 96 well plates at a cell density of 3 × 10^3^/well in 100 µL of complete RPMI (10% FBS, 1% antimycotic, and 0.6% antibiotic) and maintained at 37 °C with 5% CO_2_ humidity. Various concentrations of PG (5, 10, 20, and 50 µM) were used to treat the cells. Following treatment, 100 µL of MTT reagent was added to each well after 24, 48, and 72 h of treatment and incubated for 3 h at 37 °C. After incubation, the insoluble formazan crystals were solubilized with a 90% isopropanol (Fisher Scientific, Waltham, MA, USA) and 10% Triton-X (Sigma-Aldrich, St. Louis, MO, USA) solution. The plates were kept on the shaker for 15 min to solubilize all the crystals completely, and absorbance was measured at 590 nm using a Biotek plate reader. The cell viability percentages were calculated according to the formula given below,
$$\%\;Cell\;viabilty=\frac{Avg.\;\;OD\;of\;experimental\;cells}{Avg.\;\;OD\;of\;control\;cells}\times 100$$

Moreover, to determine the cell death mechanism, a cell viability assay was performed using a combination of varying concentrations of PG and different inhibitors such as NAC, 3MA, NEC, and CI to block the PG effect. In addition, to test cytotoxicity and hepatotoxicity of PG, well-established MPCCs of primary human hepatocytes and supportive 3T3-J2 murine embryonic fibroblasts, as well as 3T3-J2 fibroblast-only monocultures, were created as previously described [40]. All cultures were functionally stabilized for 7 days and then treated for 6 days with fresh PG added to a serum-free culture medium every 2 days. Cell viability was assessed 2 days after every PG treatment (i.e., day 9, 11, and 13 of culture) via the PrestoBlue™ cell viability assay (Invitrogen, Waltham, MA, USA).

### 2.4. Colony Formation Assay

To assess the anchorage-independent cell growth of PG on PCa cell lines, a colony formation assay was used, according to Samy et al. [41]. The colony formation assay investigates the different capacities of a single cell to form a colony in treated cells compared with the untreated control. PCa cells were seeded in a 6-well plate with complete RPMI media containing approximately 500 cells per well, and cells were allowed to grow at 37 °C with 5% CO_2_. After 24 h, the cells were treated with different concentrations of PG (5, 10, 20, and 50 µM) and incubated for 1–3 weeks. The media was changed every 3 days during incubation. To observe colony formation, the media was removed, and the cells were fixed with acetic acid and methanol in a ratio of 1:7 and incubated at room temperature for 5 min. The fixative agent was removed, and the cells were stained with 1 mL of 0.5% (*v/v*) crystal violet (Fisher Scientific, Waltham, MA, USA) and incubated for 2 h at room temperature. Then, the plates were washed with running tap water twice and allowed to dry for 5 days.

### 2.5. Wound Healing Assay

Wound healing assay is an in vitro method to study the migration ability of metastatic cancer cells. The assay evaluates the inhibitory effect of PG on cell migration characteristics in a PCa model [42]. The PCa cells were cultured in a 24-well plate and incubated at 37 °C with 5% CO_2_. After the cell reached 100% confluency, the old media was removed, and the cells were treated with mitomycin C at a concentration of 15µM for 1 h. After that, each well was washed with 1 mL of 1X PBS, then a scratch (wound) was made on the surface of each well using a 200 µL pipet tip, and the wells were washed with PBS. Complete RPMI media was added to each well and treated with different concentrations of PG (5, 10, 20, and 50 µM). The wound closing of PCa cell line image was taken using a microscope (Olympus IX73) at 0, 24, 48, and 72 h after treatment. The percentage of open area over treatment data was analyzed using T scratch image software.

### 2.6. Apoptosis Assay by Immunofluorescence Analysis

To examine whether PG induces apoptosis in PCa cell lines, an apoptosis assay was performed using annexin V FITC/PI staining as per the protocol of Gong et al. [43]. In an 8-well chamber, PCa cells were seeded at a density of 10,000 cells per well in 250 µL of complete RPMI media and allowed to grow at 37 °C with 5% CO_2_ for 48 h. The spent media were replaced with fresh complete RPMI media, and the cells were treated with different concentrations of PG (5, 10, 20, and 50 µM). After 48 h of incubation, each well was washed with 250 µL of 1X PBS and stained with 2.5 µL annexin V FITC and PI using the BD Annexin V: FITC apoptosis detection kit-I (BD Bioscience, San Jose, CA, USA). The plate was incubated in the dark at room temperature for 15 min. Again, the cells were washed with 250 µL of 1X PBS. After removing the chamber, a drop of fluorogel was added to each well and covered with a coverslip. The slides were imaged under an Olympus FV10i confocal microscope.

### 2.7. Apoptosis Assay by Flow Cytometry

To dissect the mechanism by which PG induces apoptosis in PCa cell lines, we performed flow cytometry analysis using annexin V FITC / PI staining as per the protocol of Wlodkowic et al. [44]. PCa cells were seeded at a density of 50,000 cells per well in a 6-well plate and incubated at 37 °C with 5% CO_2_ for 48 h. The cells were treated with various concentrations of PG (5, 10, 20, and 50 µM). After 48 h of treatment, both live and dead cells were collected and stained with 2.5 µL of annexin V FITC/ PI using the annexin V: FITC apoptosis detection kit I, followed by incubation in the dark for 15 min at room temperature. The treated and untreated cells were then analyzed via flow cytometry (FACS Calibur; Becton Dickinson, Mountain View, CA, USA).

### 2.8. Anchorage-Dependent Cell Death Assay

Anchorage-dependent cell death assay, also known as anoikis, is a type of apoptosis that is induced by a lack of appropriate cell-extracellular matrix (ECM) adhesion. Healthy cells undergo cell death in the absence of ECM adhesion, whereas cancer cells evade anoikis to metastasize to another organ [45]. To evaluate whether PG promotes anoikis, the assay was performed by calcein-AM and EthD-1 staining, and the MTT assay was performed as per Kummrow et al. [45], with modifications. The plates were coated with poly-HEMA to prevent the cells from adhering to the plate, and approximately 10,000 PCa cells were seeded in each well and incubated at 37 °C with 5% CO_2_. After 24 h of incubation, the cells were treated with varying concentrations of PG (1, 5, 1, 2, and 50 µM). After 24 h of treatment time, the cells were stained with calcein-AM (stains live cells) and EthD-1 (stains dead cells), observed under the immunofluorescence microscope (Olympus IX73), and images were acquired. In addition, we assessed cell viability by adding 10% MTT to each well in a 24-well anchorage resistant plate and incubating overnight as per Chinnapaka et al. [46]. The media was removed, leaving behind the purple formazan crystals, which were solubilized with a solubilizing agent. The absorbance was measured at 590 nm using a Biotek plate reader.

### 2.9. Quantitative Expression of Anchorage-Dependent Cell Death Markers by Real Time-qPCR Analysis

To confirm that PG promotes anoikis, PCa cells were treated with different concentrations of PG (5,10, 25, and 50μM) for 48 hrs. The cells were harvested, and total RNA was extracted using the TRIzol reagent (Invitrogen, Waltham, MA, USA), as per Dasari et al. [20]. The complementary DNA (cDNA) was synthesized using a high-capacity cDNA reverse transcription kit (Fisher Scientific, Waltham, MA, USA). Real-time PCR was performed using the Maxima SYBR green qPCR master mix (Fisher Scientific, Waltham, MA, USA). The expression level of β-actin, VIM, CDH1, SNAI1, SNAI2, TWIST1, and ZO-1 mRNA was quantified by real-time PCR QuantStudio 3 (Applied Biosystems, CA, USA). β-actin was used as an endogenous control. The relative gene expression was analyzed using the fold change (2^(−Δ(ΔCt))^) of treated samples.

### 2.10. Cell Cycle Analysis by Flow Cytometry

To investigate whether PG promotes cell death by inducing cell cycle arrest in the PCa cell line, flow cytometry analysis was performed. The DNA was quantitatively analyzed using the DNA binding dye PI, as per Pozarowski et al. [47]. PCa cell lines were cultured in a 6-well plate at a density of 50,000 cells per well with complete RPMI media and allowed to grow at 37 °C with 5% CO_2_ for 48 h. After 48 h of treatment with varying concentrations of PG, the cells were collected and fixed with ice-cold 70% ethanol and incubated overnight at 4 °C. The cells were washed with 1X PBS, treated with 100 µg/mL RNAase, and incubated at 37 °C for 30 min. After washing with 1X PBS, the cells were stained with 50 µg/mL PI, and the samples were incubated at room temperature for 30 min. The cells were then analyzed using flow cytometry (FACS Calibur; Becton Dickinson, Mountain View, CA, USA).

### 2.11. Reactive Oxygen Species Assay

Cancer cells exhibit a basal level of reactive oxygen species (ROS), which is a highly reactive molecule that causes proteins, lipids, and nucleic acid damage in high concentrations. Intracellular antioxidants detoxify excess ROS. ROS levels are elevated when the balance between ROS and antioxidants is disrupted, leading to oxidative stress-mediated cell death [48]. An ROS assay was performed to examine whether PG promotes ROS generation, mediating cell death in taxane-resistant CRPC cell lines, as per Casaburi et al. [49]. Approximately 10,000 cells were seeded with complete RMPI media in an 8-well chamber and incubated at 37 °C with 5% CO_2_ for 48 h. The cells were treated with different concentrations of PG. After 6 h, the cells were washed with 1X PBS and stained with 2.5 µL of the ROS fluorescent dye 2′,7′-Dichlorodihydrofluorescein diacetate (DCFH2-DA) and incubated at room temperature in the dark for 15 min. The chambers were removed from the slide, then a drop of fluorogel was added to each well and covered with a coverslip. The cells were observed under the immunofluorescent microscope, and images were acquired.

### 2.12. Reactive Oxygen Species Analysis by Flow Cytometry

To affirm our results from the previous experiment, ROS production was analyzed by flow cytometry using DCFH2-DA. Approximately 10,000 cells were seeded with complete RMPI media in a 6-well chamber and incubated at 37 °C with 5% CO_2_ for 48 h. At 70–80% confluence, cells were treated with 20 μM DCFH2-DA and incubated for 45 min. After incubation, cells were treated with different concentrations of PG for 4 h. After incubation, cells were centrifuged at 1000 rpm for 7 min and washed with 1X PBS. Fluorescence was measured using flow cytometry (FACS Calibur; Becton Dickinson, Mountain View, CA, USA).

### 2.13. Proteome Profiler-Human Apoptosis Array

To determine the expression profile of various apoptotic markers and to understand the mechanism of PG, a proteome profiler was performed using a human apoptosis array kit (R&D Systems Inc., Minneapolis, MN, USA). The PC3-TXR cells were cultured in 2 T-25 flasks and grown for 48 h at 37 °C with 5% CO_2_. One flask was treated with PG 20 µM, and the other flask remained untreated and was incubated for 12 h. Using a cell lysis buffer, cell lysates were collected from untreated and treated flasks. Capture and control antibodies were spotted on the array (nitrocellulose membrane) provided along with the kit. The collected lysates (untreated and PG 20 µM) were incubated overnight with a nitrocellulose membrane array. After incubation, the array was washed 3 times with 1X wash buffer to remove the unbound proteins, followed by 1 h incubation with a cocktail of biotinylated detection antibodies on a rocking platform. The array was washed 3 times with 1X wash buffer and incubated in streptavidin HRP on a rocking platform. Again, the array was washed thrice with 1X wash buffer, chemiluminescent reagents were added, and the signals obtained as spots were captured using X-ray films. The spot intensity was analyzed using a Dot-Blot Protein Array Analyzer macro for Image J software developed by Carpentier in 2008 [50]. The macro is accessible at (https://imagej.nih.gov/ij/macros/toolsets/Dot%20Blot%20Analyzer.txt) (accessed on 4 April 2022) [50].

### 2.14. Western Blotting

Western blot analysis was performed to determine the anticancer mechanisms of PG and to depict the underlying pathway that induces cell death [51,52]. PCa cell lines were seeded in T25 flasks and grown at 37 °C with 5% CO_2_ for 48 h. The cells were treated with varying concentrations of PG and incubated for a treatment period of 48 h. Cell lysates, protein estimation, and gel loading were performed as per Chinnapaka et al. [53]. After loading, the gels were run at 140 volts for 70 min, then transferred to the nitrocellulose membrane using the semi-dry transfer (Bio-Rad Laboratories, Inc., California, USA) at 25 volts for 60 min. The membrane was incubated with 5% skimmed milk in TBS for 1 h. After blocking, the membrane was washed thrice with tris-buffered saline tween 20 (TBST) every 10 min. The membrane was probed with respective primary antibodies and incubated overnight at 4 °C. Similarly, the blots were washed 3 times with TBST and probed with the HRP conjugated secondary antibody for 1 h. The signals were detected using the SuperSignal™ West Pico PLUS Chemiluminescent Substrate (Thermo Scientific, Waltham, MA, USA) and developed using X-ray films. The band intensities were analyzed using ImageJ software. Original blots see Appendix A.

### 2.15. Statistical Analysis

Quantitative data were analyzed using the One-way, and Two-way analysis of variance (ANOVA) test followed by Tukey’s multiple comparison test to determine the treatment significance between the PG treated and the controls within each cell line. A *p*-value ≤ 0.05 is considered to be statistically significant. GraphPad Prism version 9 was used for the statistical analysis.

## 3. Results

### 3.1. PG Is Less Toxic in Hepatocytes and 3T3-J2 Fibroblasts and Inhibits the Cell Viability of Taxane-Resistant CRPC Cell Lines in a Concentration-Dependent Manner

The cell viability assay is used to investigate the cytotoxic effect of the drug in cell lines. It is based on determining the colorimetric change resulting from the enzymatic conversion of the water-soluble MTT dye to water-insoluble formazan crystals by mitochondrial dehydrogenases in the live cells [54]. Hepatocyte/3T3-J2 fibroblast MPCCs and 3T3-J2 fibroblast-only monocultures were treated with increasing concentrations of PG. Our results show no significant fold change difference in the treated cells compared to the DMSO control (Figure 1A,B). This insinuates that PG exhibits an innocuous effect on normal cells.

Similarly, the PC3-TXR and DU145-TXR cell lines were treated with varying concentrations of PG. Our results reveal that PG significantly decreases cell viability with the increase in PG concentration, as shown in Figure 2A,B. In both the cell lines, the half maximal inhibitory concentration (IC50) is 20 µM, and the maximum inhibition of cell viability by PG is observed at 50 µM. Different time intervals (24, 48, and 72 h) show similar cell viability at higher PG concentrations. Furthermore, to study the potential mechanism of cell death caused by PG, cell viability–blocking experiments were performed. The assay was performed by several inhibitors, including 3MA, NEC, CI, and NAC. In PC3-TXR (Figure 2C), we find that NAC, CI, and 3MA significantly abrogate the cytotoxic effect of PG at a concentration of 10 µM. In addition, the cytotoxic effect of PG at a concentration of 20 µM is considerably inhibited by NAC, CI, 3MA, and NEC within 48 h of treatment.

In contrast, in DU145-TXR (Figure 2D), we observed that NAC, CI, and 3MA significantly block the cytotoxic effect of PG at a concentration of 20 µM. Moreover, we examined the impact of PG on the non-malignant BPH-1 cell line compared to cancerous cell lines PC3-TXR and DU145-TXR. Our results show that PG has an innocuous effect on BPH-1 in low concentrations (Figure 2E). Overall, our data suggest that the taxane-resistant CRPC cell lines are sensitive to PG treatment, indicating that PG has a higher potential for targeting taxane-resistant CRPC. Moreover, the effect of PG was predominately blocked by the antioxidant NAC, implying that PG induces oxidative stress as a potential anticancer mechanism.

### 3.2. PG Inhibits Colony Formation in Taxane-Resistant CRPC Cell Lines

The clonogenic assay is used to evaluate the capability of a single cell to proliferate independently and form a colony. The assay was conducted to examine the effectiveness of the cytotoxic compounds in tumor-forming (colony formation), a characteristic of cancer in in vitro conditions [55]. We observed a substantial decrease in the number of colonies compared to those in the control in both PC3-TXR and DU145-TXR cells treated with PG (Figure 3A,B). In addition, we observed that NAC significantly abrogates the PG effect in both cell lines (Figure 3C,D). Collectively, our findings demonstrate that PG substantially decreases the number of colonies in taxane-resistant CRPC cell lines in a concentration-dependent manner.

### 3.3. PG Inhibits In Vitro Migration Ability of Taxane-Resistant CRPC Cell Lines

The migration of cancer cells from the primary site to nearby tissues or distant organs is a hallmark of cancer [56]. For the cells to metastasize from one organ to another, the cells must dissociate from the primary site, enter circulatory and lymphatic systems, extravasate at distant capillaries, and invade other organs as a secondary tumor [57]. Wound healing is an inexpensive, robust, in vitro method that mimics the in vivo cell migration system. To investigate the effect of PG on migration, wound healing was performed. Our study demonstrates that taxane-resistant CRPC cell lines treated with varying concentrations of PG have a higher percentage of open wound area compared to the control when measured at different time intervals (0, 24, 48, and 72 h) (Figure 4). PC3-TXR cells treated with PG at 50 µM had 45.5% of the open area after 72 h, whereas in the control, the percentage was 0%. In the case of DU145-TXR, the percentage of the open area after 72 h was 34.7% in PG at 50 µM and 0% in the control (Figure 4D). Consistent with our results in the colony formation assay, our data also show that NAC blocks the effect of PG in both cell lines. Moreover, in both taxane-resistant CRPC cell lines, we observed that PG inhibits migration in a concentration-dependent manner. Hence, taxane-resistant CRPC cell lines are susceptible to the antimetastatic effect of PG.

### 3.4. PG Induces Programmed Cell Death in Taxane-Resistant CRPC Cell Lines

Programmed cell death, also well known as apoptosis, is a cellular process associated with maintaining the physiological balance between cell death and cell growth [58]. Cancer cells mostly evade apoptosis. The previous experiments reveal that PG inhibits cell viability, colony formation, and metastasis. We want to study whether PG induces apoptosis in taxane-resistant CRPC cell lines. Cells undergoing apoptosis evince a specific morphological change, such as loss of plasma membrane asymmetry, condensation of the nucleus and cytoplasm, and chromatin condensation. In early apoptosis, the membrane lipid molecule phosphatidylserine is flipped from the inner leaflet to the outer leaflet. To determine the apoptosis, we used annexin V conjugated to the green fluorescence dye FITC, which stains the cytoplasm, and propidium iodide (PI), which stains the DNA. Early apoptotic cells were stained with annexin V FITC (green fluorescence), whereas late apoptotic cells took both annexin V FITC (green fluorescence) and PI (red fluorescence) [44].

Figure 5A,B shows that PG induces apoptosis in PC3-TXR cells. At higher concentrations of PG (50 µM), we found a marked increase in the annexin V FITC green fluorescence, whereas, at lower concentrations of PG (5, 10, and 20 µM), we found weak, faint green fluorescence. Likewise, DU145-TXR cells treated with various concentrations of PG also showed apoptosis with a relative increase in FITC signal. PG concentrations of 20 and 50 µM showed a significant increase in green fluorescence, whereas PG 5 and 10 µM treatments displayed faint green fluorescence (Figure 5C,D). Together, these results indicate that PG induces apoptosis in taxane-resistant CRPC cell lines.

### 3.5. PG Promotes Apoptotic Cell Death in Taxane-Resistant CRPC Cell Lines

To further confirm the apoptotic-inducing potential and to quantitively examine whether PG initiates apoptosis in taxane-resistant CRPC cell lines, we assessed apoptosis using flow cytometry. The viable cells are negative for annexin V and PI stains, whereas early apoptotic cells are positive for annexin V only as plasma membrane integrity is not entirely lost; late apoptotic cells are positive for annexin V and PI. Figure 6 illustrates that taxane-resistant CRPC cell lines treated with varying concentrations of PG undergo apoptosis. PC3-TXR showed 42.32% of apoptosis (positive for annexin V and PI) at a concentration of 50 µM, whereas the control only showed 4.28% (Figure 6A,C). Likewise, in DU145-TXR, we found that the apoptosis percentage was 30.24% in treatment group and 4.59% in controls at a concentration of 50 µM (Figure 6B,D). Together, our results suggest that the percentage of apoptosis is significantly higher in treatment samples compared to that of controls, indicating that taxane-resistant CRPC cell lines are particularly susceptible to apoptosis induced by PG treatment.

### 3.6. PG Promotes Anoikis in Taxane-Resistant CRPC Cell Lines

Anoikis is a self-defense mechanism that acts as a barrier to prevent metastasis. Healthy cells undergo death when they detach or lose contact with the ECM through an apoptotic process known as anoikis. Cancer cells have the ability to escape this mechanism. The cells can expand, invade, and disseminate throughout the body, causing metastasis. Anoikis has a vital role in modulating metastasis in cancer [58,59]. Therefore, we assessed the potential role of PG in promoting anoikis in taxane-resistant PCa cell lines. We treated taxane-resistant PCa cell lines with various concentrations of PG and stained the cells with calcein-AM and EthD-1. Calcein-AM, a green fluorescence dye, stains live cells, whereas EthD-1, a red fluorescence dye, stains cells undergoing anoikis. Using the immunofluorescence data of calcein-AM and EthD-1 of control and PG-treated cells, we observed that the number of dead cells stained with EthD-1 (red fluorescence) increased in association with the increase in PG concentration.

In contrast, the number of live cells stained with calcein-AM (green fluorescence) decreased, suggesting that PG promotes anoikis in taxane-resistant CRPC cell lines (Figure 7A,B). In the MTT assay, we observed that a higher concentration of PG (20 and 50µM) exhibited a robust reduction in the percentage of cell viability in PC3-TXR (Figure 7C), while PG exhibited a marked decrease in the percentage of cell viability at a concentration of 50 µM in DU145-TXR (Figure 7D). Altogether, our results reveal that PG treatment induces anoikis in taxane-resistant CRPC cell lines.

### 3.7. PG Induces Anoikis in Taxane-Resistant CRPC Cell Lines via PTEN

Several studies have discussed the connection between PTEN and anoikis, showing that PTEN induces anoikis in several cancers [60,61,62,63]. Our previous results show that PG promotes anoikis in taxane-resistant PCa cell lines. Therefore, we examined the expression level of PTEN in taxane-resistant PCa cell lines PC3-TXR and DU145-TXR, respectively. Our data reveal that PG at a higher concentration of 20 µM significantly induces the expression of PTEN in both cell lines, indicating that PG induces anoikis in taxane-resistant CRPC cell lines (Figure 8A,B).

### 3.8. PG Causes G0 Phase Cell Cycle Arrest in Taxane-Resistant CRPC Cell Lines

We investigated the effect of PG on the cell cycle of taxane-resistant CRPC cell lines using flow cytometry to examine whether PG treatment impacts cell cycle progression. In this technique, PI binds to the DNA, and the amount of DNA present correlates to the relative intensity of PI. In PC3-TXR, G0 phase cells in the control have 4.64% of total cells, whereas G0 phase cells in the treatment sample at a concentration of 50 µM have 67.08% of cells (Figure 9A,C). Similarly, in DU145-TXR, G0 phase cells in the control have 1.46% of cells. Interestingly, treatment with PG at a concentration of 50 µM significantly increases the proportion of G0 phase cells to 88.10% (Figure 9B,D). We did not detect significant differences with other concentrations (5, 10, and 20 µM). Overall, our results suggest that PG causes G0 phase cell cycle arrest in a concentration-dependent fashion.

### 3.9. PG Treatment Induces Oxidative Stress in Taxane-Resistant CRPC Cell Lines

Cancer cells exhibit elevated ROS levels; however, the cells produce high amounts of antioxidants to counteract the generated ROS and maintain intercellular balance to abrogate the high level of oxidative stress that drives the cell to commit apoptosis. Generally, a high percentage of chemotherapy drugs trigger cell death by promoting ROS generation, which results in elevated oxidative stress and cell death [64]. To investigate whether PG treatment could result in ROS generation and, subsequently, cell death, we performed a ROS assay. The relative ROS level was determined using ImageJ software, which correlates with the intensity of green fluorescence produced. As shown in Figure 10, our data illustrate an increase in green fluorescence upon using PG treatment at a concentration of PG 20 µM compared to the control. Nonetheless, PG at a concentration of 50 µM does not show a remarkable increase in green fluorescence compared to the control, possibly due to non-viable cells at higher concentrations. Our results suggest that PG induces ROS generation in taxane-resistant CRPC cell lines and subsequently promotes oxidative stress and cell death.

Furthermore, to confirm our findings, we examined whether PG induces ROS production in taxane-resistant PCa cell lines by the means of flow cytometry. For this assay, we used the DCFH2-DA dye to detect ROS production. Our results demonstrate that in the PC3-TXR cell line (Figure 11A), PG significantly induces ROS production at a concentration of 20 µM. Similar results are also observed in DU145-TXR (Figure 11B). Moreover, our data show that NAC markedly blocks the effect of PG in both cell lines (Figure 11C,D). Collectively, our results indicate that PG promotes ROS production and consequently induces oxidative stress and cell death.

### 3.10. PG Differentially Modulates the Expression of Various Apoptotic Markers in PC3-TXR

The previous experiments revealed that PG induces apoptosis in PC3-TXR cells. As a follow-up to previous investigations demonstrating PG apoptotic-inducing potential, we wanted to examine the specific expression of various apoptotic markers by an apoptosis profiler array, which is a rapid method to detect the expression of 35 apoptosis-related proteins in a single array (Figure 12). The principle behind this experiment remains the same as in a Western blot assay. For apoptosis marker profiling, we chose PC3-TXR cells as both PC3-TXR and DU145-TXR cell lines showed similar results in previous experiments. Proteome array results illustrated that PG modulates key apoptosis markers. Using PG at a concentration of 20 µM, our data show downregulation in the expression level of antiapoptotic proteins Bcl-2 and Bcl-xL. In addition, the expression level of the inhibitors of apoptosis (IAP) proteins cIAP-1, cIAP-2, survivin, and XIAP expressed downregulation as well. The IAP family endogenously inhibits apoptosis by binding to the caspases, abrogating programmed cell death, and enhancing tumor cell proliferation [65,66].

Furthermore, our results also show that the expression level of cytochrome c is elevated. In addition, pro-caspase-3 is downregulated, and cleaved caspase-3 is upregulated, indicating that PG induces cell death via the activation of the intrinsic apoptosis pathway. Cells undergo apoptosis via the intrinsic pathway characterized by the disruption of mitochondrial function, resulting in the release of cytochrome c and eventually caspase activation. Caspase-3 is an executioner among the caspases in programmed cell death [67]. In summary, our results suggest that PG treatment activates the intrinsic apoptosis pathway in CRPC cells.

#### PG Regulates the Expression of Key Apoptotic Markers and DNA Damage Markers in Taxane-Resistant CRPC Cell Lines

Based on the human apoptosis array results, it is evident that PG treatment modulates various apoptosis markers in PC3-TXR. To further study the mechanism of PG action, immunoblot analysis was conducted to substantiate the expression of apoptotic and DNA damage proteins. Immunoblot results (Figure 13) illustrate that PG treatment upregulates the expression of DNA damage response protein pH2AX in taxane-resistant CRPC cell lines. In addition, we also observed downregulation of PARP-1, pro-caspase-3, XIAP, and cIAP-2 in both cell lines, implicating apoptosis activation [68]. Moreover, to confirm that PG induces the cell to undergo apoptosis, we examined the expression of cleaved PARP-1 in both cell lines. Our data show that cleaved PARP-1 expression is upregulated while total PARP-1 is downregulated, indicating that cells commit apoptosis (Figure 13E,F). The expression of β-actin was observed to be homogeneous in treated as well as untreated samples. Taken together, our work shows that PG treatment causes apoptosis-mediated cell death.

## 4. Discussion

PCa is the most frequent cancer among males in the United States. Surgery, chemotherapy, radiation therapy, and ADT are the currently available treatment options for PCa. Although there have been recent advancements in treatment strategies, challenges still obstruct the chances of curing PCa. One of these challenges is the development of resistance to ADT therapy and, consequently, the development of CRPC after 18–24 months [14,15]. In this study, we wanted to explore the potential of PG as an alternative and natural therapeutic anticancer agent against taxane-resistant CRPC. Our results show that PG is not toxic to primary human hepatocytes, 3T3-J2 fibroblast co-cultures, and non-cancerous BPH-1 cell lines, implicating that PG is innocuous to healthy cells. Importantly, PG significantly inhibits the cell viability of taxane-resistant CRPC cell lines in a concentration-dependent manner at 24 h and 48 h time intervals. Our previous work showed that DR-P27, a derivative of PG, had antiproliferative properties against several cancer types, including apoptosis-resistant human glioblastoma U373, human SKMEL-28 melanoma, apoptosis-sensitive human Hs683 anaplastic oligodendroglioma, human A549 non-small cell lung cancer, and human MCF-7 breast cancer [30]. Similarly, our current study affirms that PG inhibits the proliferation of taxane-resistant CRPC cell lines.

Colony formation and metastasis are essential properties for tumor survival and progression. A study on piperine, an alkaloid from black pepper, revealed that piperine inhibited colon cancer’s colony formation ability in the HT-29 cell line and the growth of colon cancer spheroids [69]. Besides, Zhang et al. [70] substantiated that piperine attenuates cell migration of HOS and U2OS osteosarcoma cell lines. PG is a natural compound present in water pepper, Dorrigo pepper, and mountain pepper. Our study demonstrates high efficacy in reducing the colony size as well as the number of colonies proportionally to an increase in the concentration of PG in taxane-resistant CRPC cell lines. In addition, our wound healing data demonstrate a significant decrease in the gap closure percentage of taxane-resistant CRPC cell lines in a concentration-dependent manner, and at different time intervals compared to the control. Moreover, our data show that NAC abrogates the effect of PG, indicating that PG induces cell death by promoting oxidative stress and ROS generation. Our work suggests that PG has an antimetastatic potential impact against taxane-resistant CRPC cell lines.

Our previous work has indicated that a PG analog DR-P27 induces apoptosis in LNCaP PCa cells [28]. In this study, we examined the anticancer properties of PG by using various cellular assays. Our work demonstrates that PG induces apoptosis in taxane-resistant CRPC cell lines, and we found a significant increase in the apoptosis percentage upon using PG at a concentration of 50 µM. Moreover, anoikis is a known form of programmed cell death that ensues when a cell detaches from the ECM [58]. Cancerous cells are resistant to this mechanism, which helps in metastatic dissemination. Our data show that PG induces anoikis in taxane-resistant PCa cell lines (Figure 7).

Moreover, PTEN has been shown to induce anoikis in various cancers. To affirm our finding, we examined the gene expression of *PTEN*, and our results demonstrate that PG treatment expedites anoikis in taxane-resistant CRPC cell lines through PTEN activation (Figure 8). Similarly, another study showed that curcumin, a primary compound from the spice turmeric, promoted anoikis in non-small lung cancer [71], indicating that natural products could be a valuable strategy to vanquish the cancer resistance dilemma.

The cell cycle checkpoints do not allow cells with DNA damage to duplicate or divide; instead, the cells undergo apoptosis. However, mutated cells escape different checkpoints, resulting in uncontrolled cell growth and, subsequently, cancer development. Piperine has been shown to induce cell cycle arrest at the G0/G1 phase and apoptosis in melanoma cells [72]. Our results reveal that PG instigates G0 phase arrest at a concentration of 50 µM, indicating that PG exterminates taxane-resistant CRPC via apoptosis. In contrast, De La Chapa et al. [32] demonstrated that PG induces cell cycle arrest in the S phase in OSCC, indicating that PG might elicit different mechanisms according to the cancer type [32]. To further investigate the underlying mechanism of cell death, we have examined ROS production in taxane-resistant CRPC. Cancer cells express elevated levels of ROS; however, antioxidants present in the cancer cells detoxify ROS. This promotes cell growth progression and development. When the intercellular ROS and antioxidant balance is disrupted, the intracellular ROS threshold level increases, leading to oxidative stress–mediating apoptotic death. Our recent study has reported that DR-P27, a derivative of PG, induced ROS generation, leading to oxidative stress–mediated cell death of androgen-sensitive human PCa cells [28]. In this study, we found that PG gradually increased ROS generation from 5 to 20 µM.

Interestingly, a higher concentration of PG (50 µM) significantly increased cell death. However, we observed relatively low ROS generation. The lower levels of ROS detected at 50 µM PG treatment suggest the possibility of rapid ROS generation earlier than 6 h of treatment, resulting in ROS exhaustion. In addition, our results reveal that NAC blocks the effect of PG in both cell lines. Based on our data, we suggest that PG induces robust ROS generation, leading to oxidative stress in taxane-resistant CRPC cell lines. Previous studies have shown that the compound alantolactone, a plant-derived sesquiterpene lactone, activates apoptosis via ROS generation leading to the disruption of mitochondrial membrane potential, the release of apoptotic factor cytochrome c, the downregulation of antiapoptotic proteins, and the activation of apoptosis executioner caspase-3 in glioblastoma cells [73].

We wanted to confirm that PG induces apoptosis; therefore, we used a proteome profiler-human apoptosis array with several spotted apoptotic markers to ascertain the mechanism of action of PG in taxane-resistant CRPC. This study used only PC3-TXR cells as both the taxane-resistant CRPC cell lines exhibited indistinguishable potency with PG treatment. Apoptotic proteome array analysis revealed an increase in cytochrome c levels. Here, we postulate that PG treatment induces ROS generation leading to disruption of mitochondrial membrane potential and further activating cytochrome c release, subsequently activating different caspases.

Tumor cells depend on the Bcl-2 family of proteins to protect them from stress-induced apoptotic death. Bcl-2 and Bcl-xL are antiapoptotic proteins and essential regulators of apoptosis [65]. In our study, the proteome profiler-human apoptosis array also reveals the expression of Bcl-2 and Bcl-xL, along with the downregulated expression of the IAP group of antiapoptotic proteins such as XIAP, survivin, cIAP-1, and cIAP-2. These results substantiate that PG induces apoptosis. Since PG treatment stimulates cytochrome c upregulation, we believe this leads to cell death through the intrinsic apoptosis pathway via mitochondrial depolarization. We further investigated the expression of caspase-3 and found that pro-caspase-3 was downregulated in Western blot results, while cleaved caspase-3 was upregulated in proteome array results in PC3-TXR cells. Moreover, the expression levels of the antiapoptotic markers cIAP-2 and XIAP were also downregulated.

DR-P27 induced ROS generation and apoptosis along with the cleavage of PARP-1 and the activation of γH2AX, leading us to conclude that DR-P27 induces DNA damage response in androgen-sensitive human PCa cells [28]. Our study found that PG treatment downregulates PARP-1 expression implicating apoptosis activation. Further, in Western blot analysis, we observed an upregulation of pH2AX, another crucial DNA damage response marker associated with apoptosis or cell cycle arrest, suggesting that PG induces DNA damage response in taxane-resistant CRPC cell lines. Collectively, these results confirm our hypothesis that PG induces ROS generation leading to the disruption of the mitochondrial membrane and the upregulation of cytochrome c, followed by apoptosis/anoikis and the inhibition of various antiapoptotic factors. In addition, our data suggest that PG effectively targets taxane-resistant CRPC cell lines by activating apoptotic cell death and inhibiting antiapoptotic signaling, suggesting that PG endows similar efficacy to DR-P27 in drug resistant cancer in vitro models.

Moreover, PG also showed significant antiproliferative potency comparable to DR-P27 in Cal27-derived tumors in a xenograft model of athymic nude mice [32]. However, this is the only study that addresses PG antitumor efficacy using an in vivo model, and further investigations are warranted. Figure 14 highlights the proposed anticancer mechanism of PG against taxane-resistant CRPC cell lines.

## 5. Conclusions and Future Perspective

This study confirms that PG has promising therapeutic potential in taxane-resistant CRPC cell lines. PG effectively inhibits the cell viability, cell cycle progression, and migration properties of CRPC cells, suggesting that PG endows tumor growth suppression and metastasis inhibition potential. Furthermore, PG induces ROS generation, disrupting the mitochondrial membrane and upregulating cytochrome c, which activates the intrinsic death pathway and anoikis. The mechanistic study confirmed that PG induces DNA damage response and apoptosis in the taxane-resistant CRPC cell lines. However, additional work is needed to unveil the detailed anticancer mechanism of PG, and further in vivo studies are warranted to ascertain the therapeutic or chemopreventive usefulness of PG in managing CRPC.

## Figures and Tables

**Figure 1 cancers-14-05260-f001:**
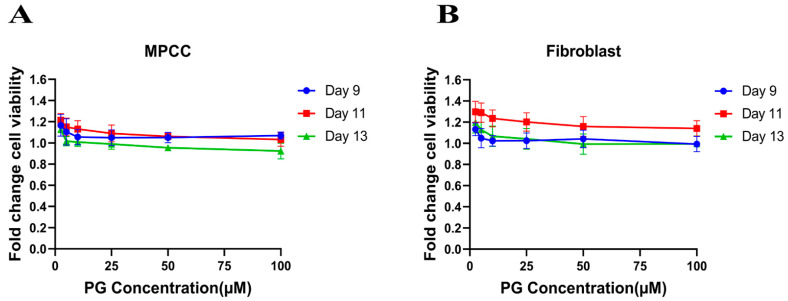
Polygodial (PG) treatments are not toxic to hepatocytes and fibroblasts. MPCCs and 3T3-J2 fibroblast-only monocultures were cultured for 7 days. Cells were treated with various concentrations of PG every other day for 6 days (i.e., three total treatments), and cell viability was measured on the 9th, 11th, and 13th day of culture. The results show that the cell viability of MPCC (**A**) and fibroblast (**B**) is not significantly affected by PG treatment.

**Figure 2 cancers-14-05260-f002:**
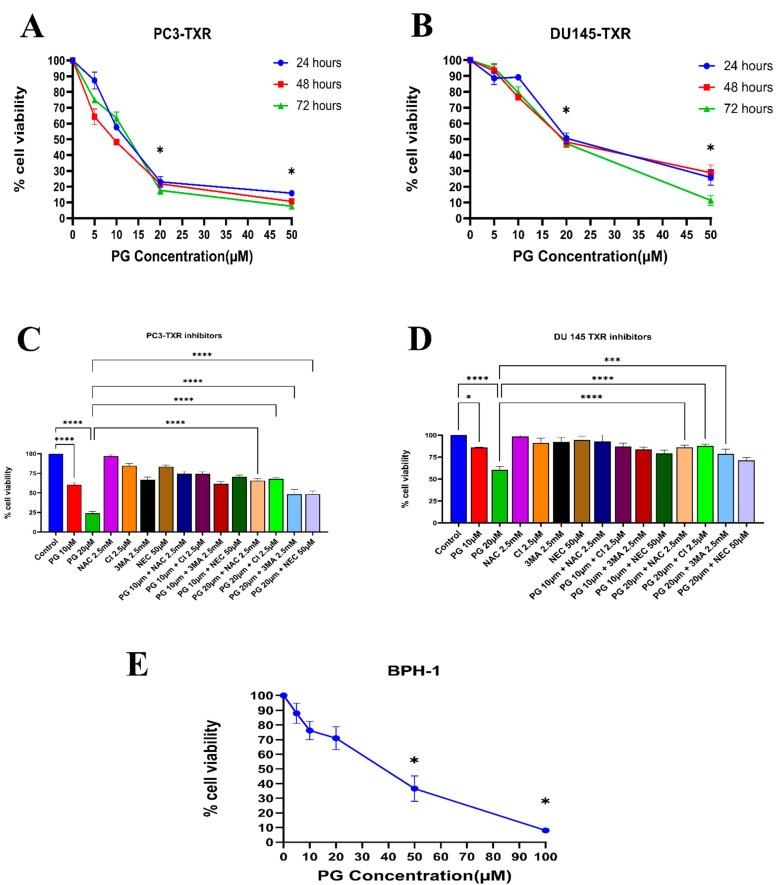
PG inhibits the cell viability of taxane-resistant PCa cell lines in a concentration-dependent manner. The taxane-resistant PCa cell lines (**A**) PC3-TXR and (**B**) DU145-TXR were treated with various concentrations of PG (treatment periods 24, 48, and 72 h). Viability was determined by MTT assay. (**C**) PC3-TXR and (**D**) DU145-TXR were treated with different inhibitors for 48 h. (**E**) depicts the effect of PG on the BPH-1 cell line. Data are represented as mean ± SD (*n* = 3) and describe three independent experiments performed in triplicate. * *p* < 0.05, *** *p* < 0.01, ***** p*< 0.001.

**Figure 3 cancers-14-05260-f003:**
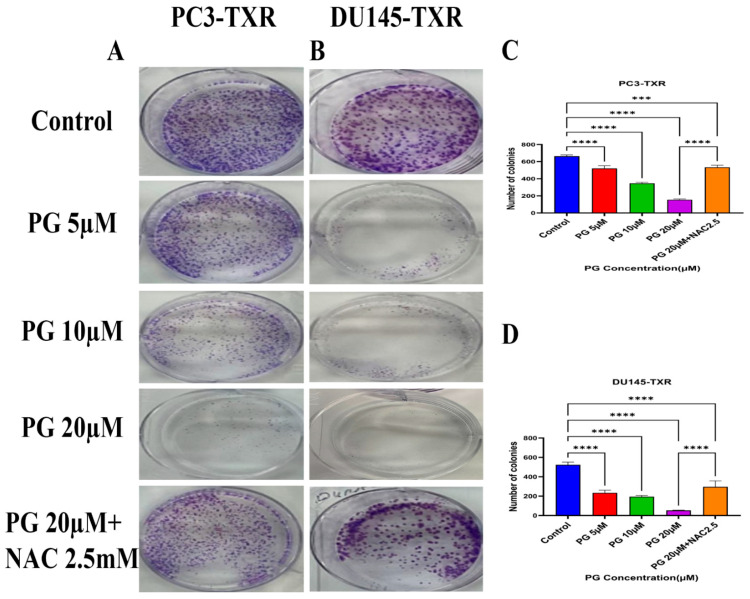
PG inhibits colony formation in taxane-resistant PCa cell lines. (**A**) PC3-TXR and (**B**) DU145-TXR represent images of colony formation after PG treatment. Compared to the control, a substantial decrease in the colonies’ number is observed with PG treatment. (**C**) and (**D**) display the quantified results of the average number of colonies plotted against varying concentrations of PG. Data are represented as mean ± SD (*n* = 3) and represent three trials performed in triplicate independently. *** *p* < 0.01, and **** *p* < 0.001.

**Figure 4 cancers-14-05260-f004:**
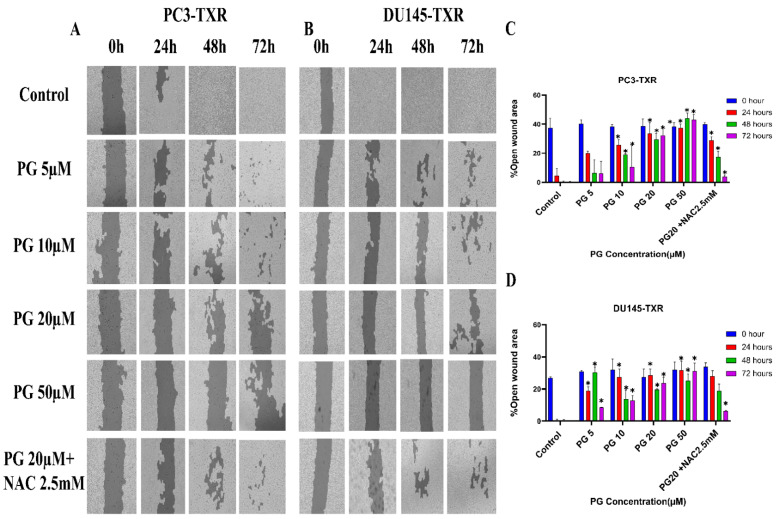
PG inhibits cell migration in taxane-resistant PCa cell lines. (**A**) PC3-TXR and (**B**) DU145-TXR cell line images were taken at 0, 24, 48, and 72 h and analyzed using T-scratch software. (**C**) and (**D**) represent the % open area plotted against varying concentrations of PG for PC3-TXR and DU145-TXR. Images were taken at 10× magnification. Data are represented as mean ± SD (*n* = 3), and three independent trials were conducted in triplicate. * *p* < 0.05.

**Figure 5 cancers-14-05260-f005:**
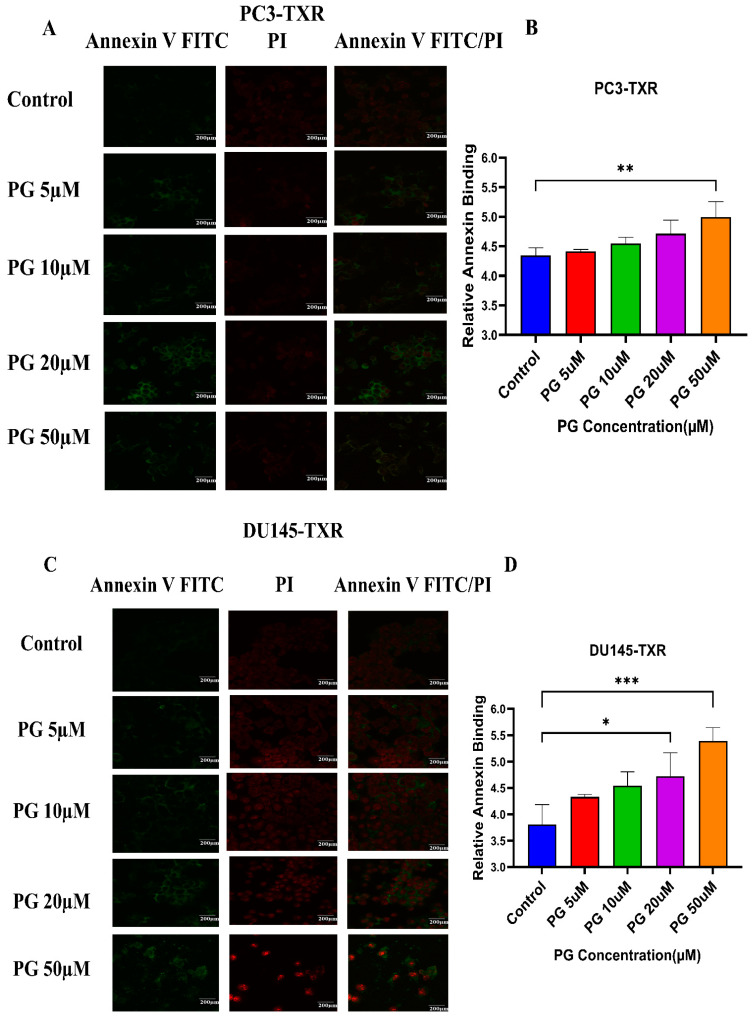
PG treatment elicits apoptosis in taxane-resistant PCa cell lines. (**A**) PC3-TXR and (**C**) DU145-TXR cell lines were treated with varying concentrations of PG for 48 h, and apoptosis was detected by annexin V FITC/PI staining. Images were taken via confocal microscopy and analyzed using ImageJ. In the taxane-resistant PCa cell lines, the relative annexin V FITC binding was higher in the treatment groups compared to that of controls. (**B**) and (**D**) show relative annexin V binding plotted against concentrations of PG for PC3-TXR and DU145-TXR, respectively. Images were taken at 60× magnification. Data are depicted as mean ± SD (*n* = 3) and represent three experiments performed in triplicate. * *p* < 0.05, ** *p* < 0.01, *** *p* < 0.001.

**Figure 6 cancers-14-05260-f006:**
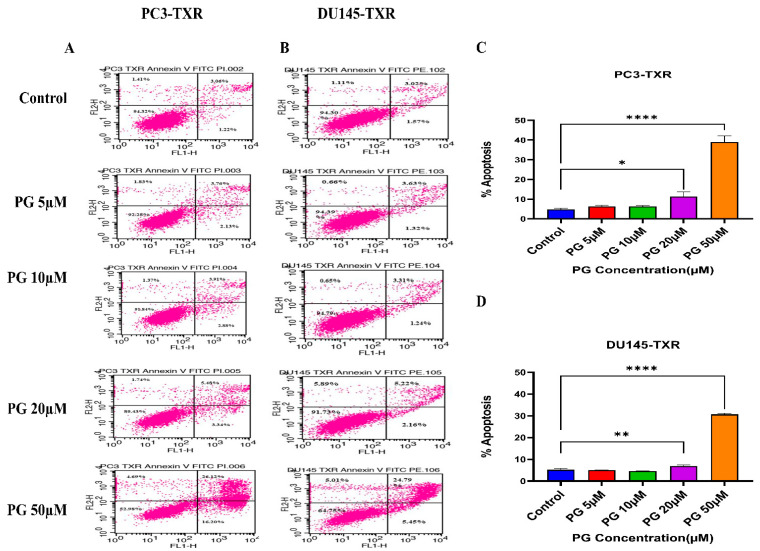
PG induces apoptosis in taxane-resistant PCa cell lines. (**A**) PC3-TXR and (**B**) DU145-TXR cell lines were treated with varying concentrations of PG, and apoptosis was detected by staining cells with annexin V FITC/PI followed by flow cytometry analysis. The results suggest that PG induces apoptosis in higher concentrations. (**C**) and (**D**) represent % apoptosis plotted against concentrations of PG for PC3-TXR and DU145-TXR. Data are shown as mean ± SD (*n* = 3) and denote three independent experiments in triplicate. * *p* < 0.05, ** *p* < 0.01, **** *p* < 0.001.

**Figure 7 cancers-14-05260-f007:**
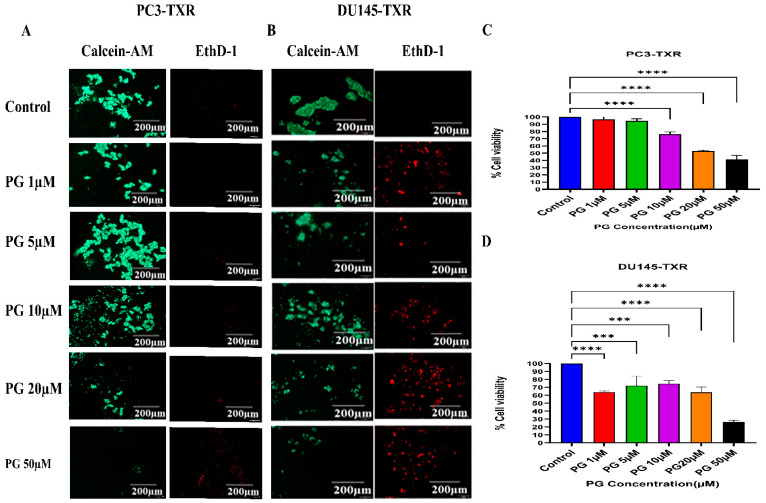
PG promotes anoikis in taxane-resistant PCa cell lines. (**A**) PC3-TXR and (**B)** DU145-TXR cell lines were treated with varying concentrations of PG for 48 h and stained with calcein AM and EthD-1 to detect anoikis. We observed that with the increase in the concentration of PG, the number of viable cells stained with calcein-AM (green fluorescence) decreased, and the number of dead cells stained with EthD-1 (red fluorescence) increased. (**C**) and (**D**) % cell viability plotted against various concentrations of PG for PC3-TXR and DU145-TXR, respectively. Images were taken at 10× magnification. Data are represented as mean ± SD (*n* = 3), and a total of three experiments were performed independently in triplicate. *** *p* < 0.05, **** *p* < 0.01.

**Figure 8 cancers-14-05260-f008:**
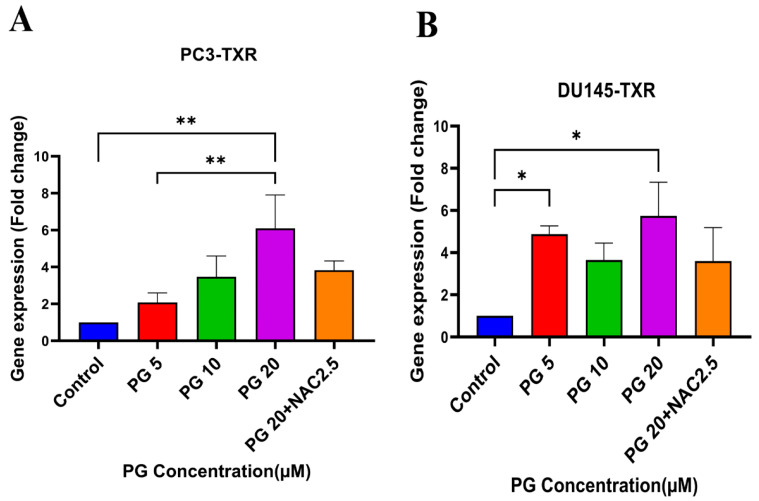
PG promotes anoikis in taxane-resistant PCa cell lines via the activation of PTEN. (**A**) PC3-TXR and (**B**) DU145-TXR cell lines. Our data illustrate that PG induces upregulation in the expression of *PTEN*. Data are represented as mean ± SD (*n* = 3), and a total of three experiments were performed independently in triplicate. * *p* < 0.05, ** *p* < 0.01.

**Figure 9 cancers-14-05260-f009:**
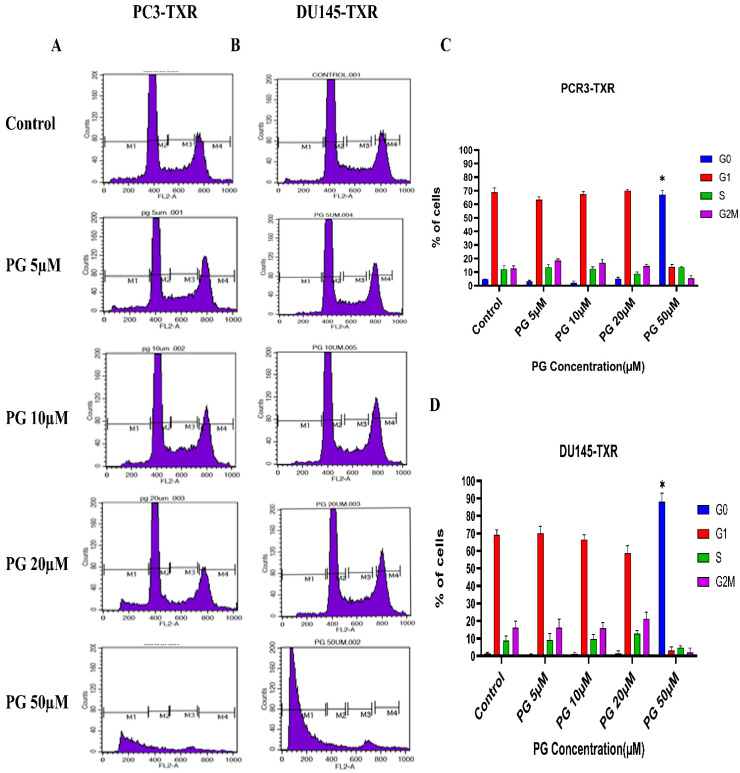
PG induces G0 phase cell cycle arrest in taxane-resistant PCa cell lines. (**A**) PC3-TXR and (**B**) DU145-TXR cell lines were treated with varying concentrations of PG for 48 h and stained with PI, and flow cytometry analysis was performed. We observed that PG 50 µM significantly blocks the cell cycle at the G0 phase in taxane-resistant PCa cell lines. (**C**) and (**D**) represent the quantified results of % of cells plotted against varying PG treatment of PC3-TXR and DU145-TXR. Data are represented as mean ± SD (*n* = 3), and three experiments were performed in triplicate. * *p* < 0.05.

**Figure 10 cancers-14-05260-f010:**
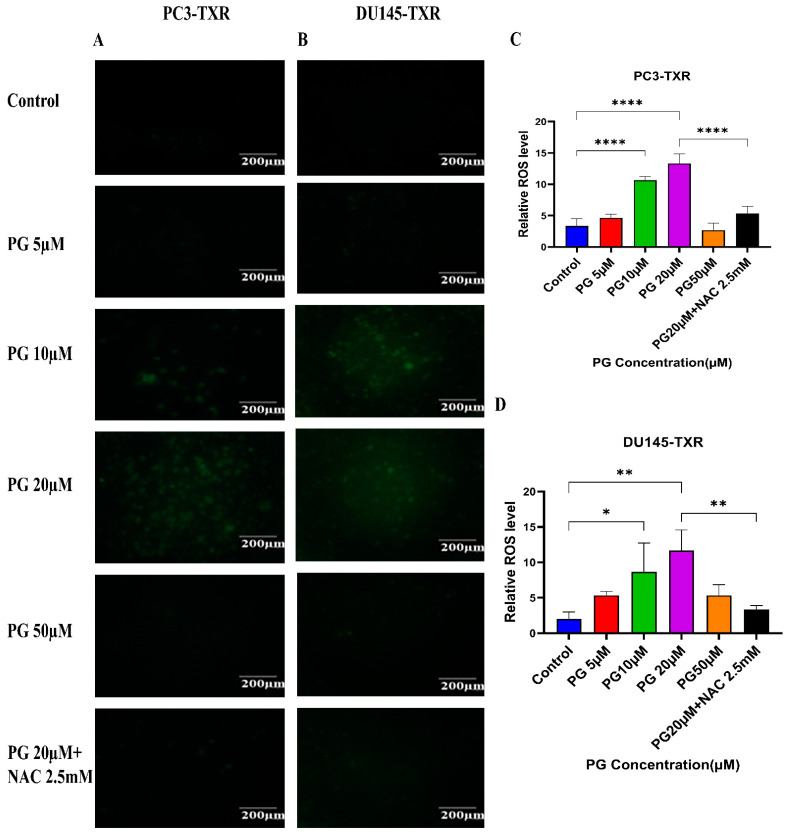
PG treatment induces ROS generation, leading to oxidative stress in taxane-resistant PCa cell lines. The taxane-resistant PCa cell lines were treated with varying concentrations of PG for 6 h and stained with DCFH2-DA. The images were taken and analyzed using ImageJ software. (**A**) PC3-TXR and (**B**) DU145-TXR show an increase in green fluorescence with an increase in PG treatment up to PG 20 µM compared to the control, but PG 50 µM shows slight green fluorescence. (**C**),**D**) represent the quantified results of relative ROS levels plotted against concentrations of PG of PC3-TXR and DU145-TXR. Images were taken at 40× magnification. Data are represented as mean ± SD (*n* = 3), and three experiments were carried out in triplicate. * *p* < 0.05, ** *p* < 0.01, **** *p* < 0.001.

**Figure 11 cancers-14-05260-f011:**
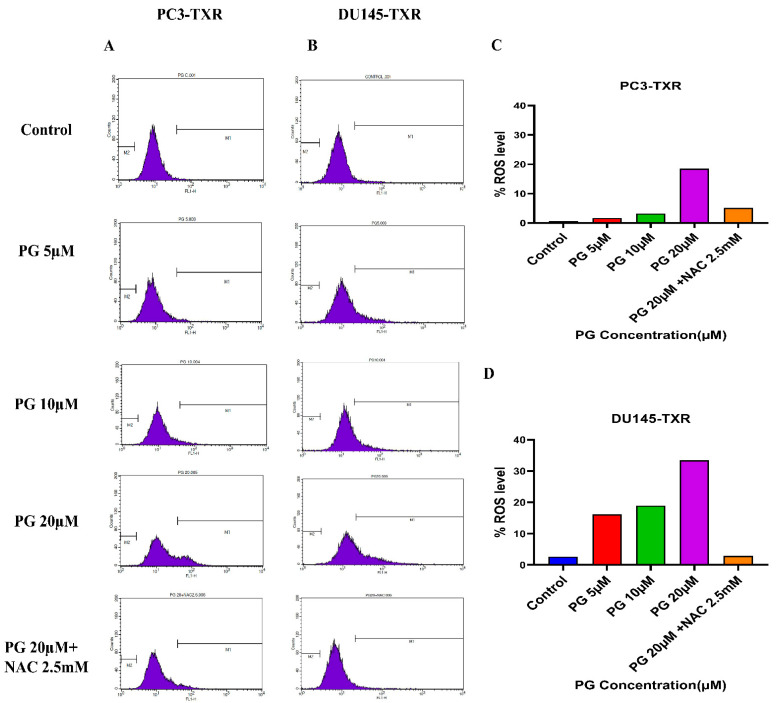
PG induces ROS production, resulting in oxidative stress in CRPC cells. PC3-TXR and DU145-TXR cells were stained with DCFH2-DA and then treated with different concentrations of PG for 4 h, followed by flow cytometry analysis. (**A**) PC3-TXR and (**B**) DU145-TXR show an increase in the amount of ROS generated with PG treatment compared to the control. (**C**) and (**D**) represent the quantified ROS levels’ results plotted against PG concentrations of PC3-TXR and DU145-TXR compared to the untreated control.

**Figure 12 cancers-14-05260-f012:**
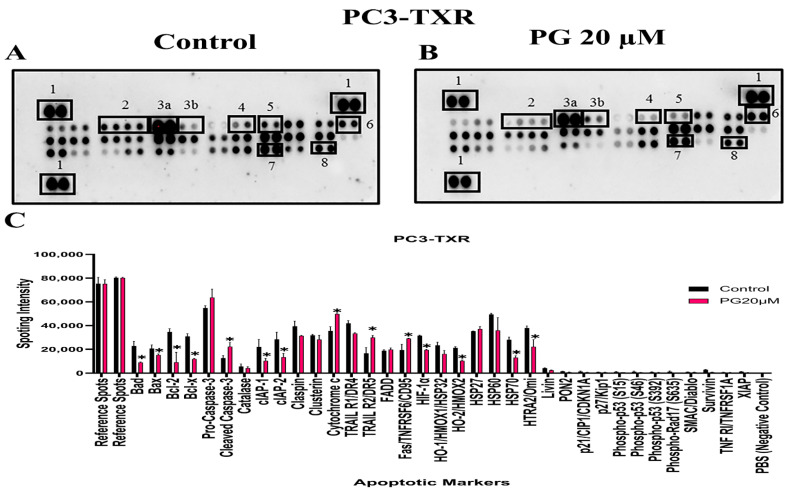
PG modulates the expression of various apoptotic markers in PC3-TXR. Figure (**A**) represents PC3-TXR control and (**B**) represents PC3-TXR treated with PG 20µM, in which the cells were seeded and treated with PG 20 µM for 48 h. Expression of the various apoptotic markers was found using a proteome profiler-human apoptosis array. PG promotes downregulation in the antiapoptotic markers Bcl-2, Bcl-xL, and IAP family, in addition to an upregulation in cytochrome c and cleaved-caspase-3. Numbers used to depict the expression of selected markers compared to reference spots: 1 = reference spot, 2 = Bcl family, 3a = pro-caspase 3, 3b = cleaved caspase 3, 4,5,8 = IAP family, 6 = cytochrome c, 7 = survivin. (**C**) represents the quantified results of the relative band intensities calculated using ImageJ, and relative expression levels were plotted against PG treatment. Data are represented as mean ± SD. * *p* < 0.05.

**Figure 13 cancers-14-05260-f013:**
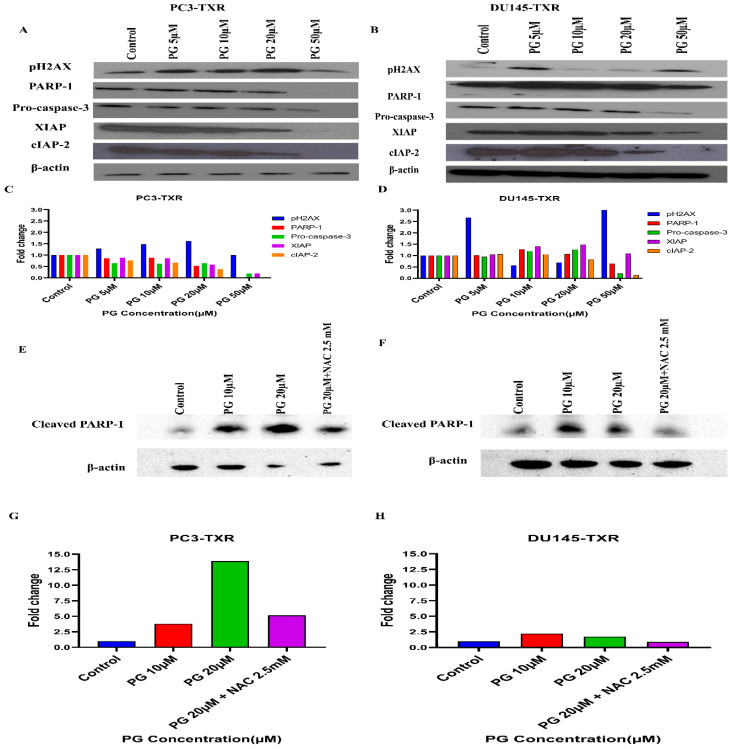
PG differentially modulates the expression of key apoptotic markers and DNA damage markers in taxane-resistant PCa cell lines. (**A**) PC3-TXR and (**B**) DU145-TXR cells were treated with varying concentrations of PG, and an immunoblot assay was performed. PG induces downregulation of various apoptotic markers, such as pro-caspase-3, XIAP, and cIAP-2. We also observed that the DNA damage marker pH2AX is upregulated, and PARP-1 is downregulated. (**C**) and (**D**) represent the quantified results of relative band intensities calculated using ImageJ, and relative expression levels were plotted against PG treatment. (**E**) and (**F**) depict the expression of cleaved PARP-1 in PC3-TXR and DU145-TXR, respectively, confirming that PG induces taxane-resistant PCa cell lines to commit apoptosis. (**G**) and (**H**) are graphical representations of the relative band intensities calculated using ImageJ, and fold change in expression levels were plotted against various PG treatments.

**Figure 14 cancers-14-05260-f014:**
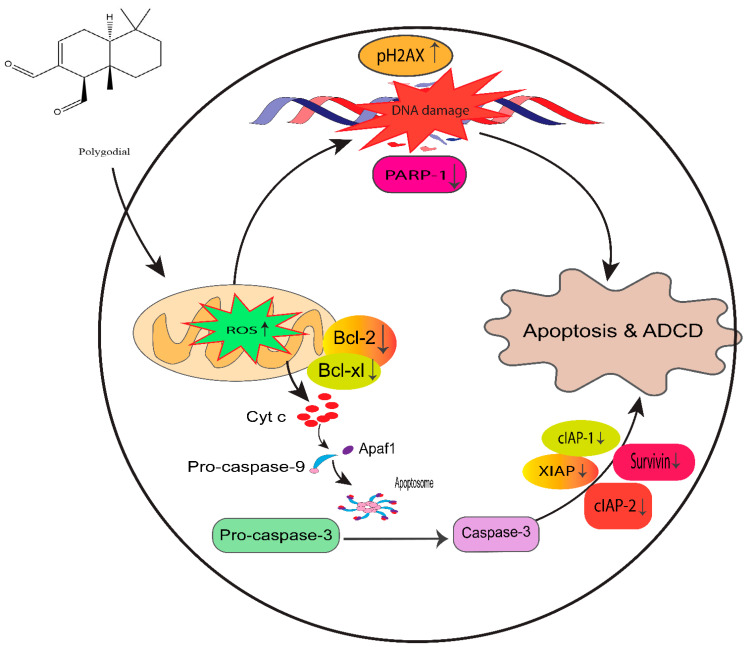
Proposed mechanism of PG action in taxane-resistant CRPC cell lines. We identified that ROS generation by PG induces DNA damage and apoptosis by the intrinsic signaling apoptosis pathway, indicating that PG has a potential therapeutic effect on taxane-resistant PCa.

## Data Availability

The data represented in this article are available on request from the corresponding author.

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
