# Peer review of "Polygodial, a Sesquiterpene Dialdehyde, Activates Apoptotic Signaling in Castration-Resistant Prostate Cancer Cell Lines by Inducing Oxidative Stress"

_cancers, 2022, doi:10.3390/cancers14215260_

Round 1

Reviewer 1 Report (Previous Reviewer 1)

Authors gave a sincere response to reply my concerns. Although some of experiments that I required authors could not complete, they provided some previous published data as evidence. Meanwhile, they also tried to improve their data. I have no more questions.

Author Response

Reviewer 1

Authors gave a sincere response to reply to my concerns. Although some of experiments that I required authors could not complete, they provided some previous published data as evidence. Meanwhile, they also tried to improve their data. I have no more questions.

  • We would like to thank the reviewer for providing us feedback on the revised manuscript and supporting our work.

Reviewer 2 Report (Previous Reviewer 2)

Generally, the article is very sloppy in terms of graphics. The figures are illegible and poorly reflect the obtained results. I understand that the preparation of a scientific paper is not an art competition, but attention to the presentation of the results may also prove how diligently they were prepared while working in the laboratory.

The article, although it comes from the USA, has several stylistic errors e.g. "Viability was determined by cell viability assay (line 357)" therefore linguistic correction is recommended.

In my opinion, the Authors should start by correcting the figures because, without correctly prepared figures, the paper cannot be adequately assessed.

Fig 1 The graphs are small and the legend shows days shifted incomprehensibly to the right. The X-axis may end at 100uM. But the biggest problem is the fold of induction on the Y axis. What value on Y-axis the X-axis = 0? It looks like it's top 1.2. Therefore, why is the "fold of induction" for 0 for the X axis around 1.2?  What's a "fold of induction" equal to 1?

Fig 2 Again, small unreadable charts. How were the IC50 values ​​from Graphs A and B calculated? Charts C and D should be better prepared, and first, at all legible, it isn't easy to deduce anything from them. There are no incubation times in the legend, charts C and D should include names and abbreviations of the inhibitors used

Fig 3 The C and D charts have unequal sizes and, what is worse, the scales on the Y axis. Moreover, it is clearly seen that the number of colonies in the DU145TXR control is greater than in the PCTXR control, the same is in the cells treated with PG20uM + NAC2.5mM while in the plots the values ​​say otherwise. PCTXR control - approximately 680 colonies (according to the scale of the chart) and DU145TXR control about 400 colonies (according to the scale of the chart). It is not known how the colonies could be counted in the DU145 TXR as they merge. The situation is similar in PG20uM + NAC2.5mM. Obviously, the data from the photos do not agree with the data from the charts. The plates like the control for DU145 simply should not be used. Especially as a control

Fig 4 Statistical analysis is missing in the graphs, although it is described in the legend. The Y axes have different scales

Fig 5 Images illegible and practically useless, it is not known how the "relative annexin binding" shown in graphs B and D was counted, and it is not known how long the cells were incubated with the tested compounds

Fig 6 The graphs in the flow cytometer are small and the resolution is low. The results do not correspond at all with the photos taken under the fluorescence microscope and the cell viability tests. In the graphs of Figure 6 20uM has minimal effect on the test cells, while in Figures 1 and 5 the changes at 20uM are very significant.

Fig 7 Again, very bad resolution, it is not known what is shown in these photos, whether they are whole cells or only parts of their structure. How do the cell viability values ​​in plots C and D correspond to the cell viability values ​​in figure 1? How long were cells with test compounds incubated?

Fig 9 The histograms are unreadable, and the graphs do not show the results of the statistical analysis, although the legend says that it was performed.

Fig 10 Pictures are illegible and virtually useless, it is not known how the "relative ROS level" shown in graphs B and D were counted, and it is not known how long the cells were incubated with the test compounds

Fig 11 Histograms are unreadable, C and D plots have unequal scales on the Y-axis, no SD, no statistical analysis performed.

Fig 12 Some of the spots are overexposed e.g. 3a. Therefore their quantitative analysis may be unreliable; no statistical analysis in the charts and in the description of the statistical analysis in the legend.

Fig 153 no SD and statistical analysis in the plots, and the scale of the Y axis in plots C, D, G, H is different, it seems that the experiments shown in the plots in Figs 12 and 13 were not performed in a sufficient number of repetitions

Author Response

Generally, the article is very sloppy in terms of graphics. The figures are illegible and poorly reflect the obtained results. I understand that the preparation of a scientific paper is not an art competition, but attention to the presentation of the results may also prove how diligently they were prepared while working in the laboratory.

  1. The article, although it comes from the USA, has several stylistic errors e.g., "Viability was determined by cell viability assay (line 357)" therefore linguistic correction is recommended.

  • As recommended by the reviewer, we have corrected the linguistic errors across the manuscript.

  1. In my opinion, the Authors should start by correcting the figures because, without correctly prepared figures, the paper cannot be adequately assessed.

  • As suggested, we have improved the size and figure resolution across the manuscript.

  1. Fig 1 The graphs are small, and the legend shows days shifted incomprehensibly to the right. The X-axis may end at 100uM. But the biggest problem is the fold of induction on the Y axis. What value on Y-axis the X-axis = 0? It looks like it's top 1.2. Therefore, why is the "fold of induction" for 0 for the X axis around 1.2?  What's a "fold of induction" equal to 1?

  • In response to the reviewer's comment, we have improved the size and figure resolution, and adjusted the legend, for the fold induction it was compared to DMSO control (Fold change= 1) and PG concentration in the X-axis corresponds to the fold change in Y-axis (Page 8, line 340 to 341).

  1. Fig 2 Again, small unreadable charts. How were the IC50 values ​​from Graphs A and B calculated? Charts C and D should be better prepared, and first, at all legible, it isn't easy to deduce anything from them. There are no incubation times in the legend, charts C and D should include names and abbreviations of the inhibitors used.

  • To address this concern, we have improved the size and resolution of figure 2. The IC50 was determined using GraphPad prism version (9) by calculating the fit dose response curve using non-linear regression. Charts C and D were updated accordingly, and the incubation time is now indicated in the figure legend (Page 9, lines 367 to 368, and page10, line 372).

  1. Fig 3 The C and D charts have unequal sizes and, what is worse, the scales on the Y axis. Moreover, it is clearly seen that the number of colonies in the DU145TXR control is greater than in the PCTXR control, the same is in the cells treated with PG20uM + NAC2.5mM while in the plots the values ​​say otherwise. PCTXR control - approximately 680 colonies (according to the scale of the chart) and DU145TXR control about 400 colonies (according to the scale of the chart). It is not known how the colonies could be counted in the DU145 TXR as they merge. The situation is similar in PG20uM + NAC2.5mM. Obviously, the data from the photos do not agree with the data from the charts. The plates like the control for DU145 simply should not be used. Especially as a control

  • To address this comment, we have revised the figure 3. The scales were adjusted, and the control DU145 TXR is updated . To clarify the comment on colony count, it was done manually and confirmed with ImageJ software using the count particle function (Page10, line 386).

  1. Fig 4 Statistical analysis is missing in the graphs, although it is described in the legend. The Y axes have different scales

  • To response to the reviewer's comment, we have updated figure 4. The scales were adjusted, and the statistics was updated (Page11, line 410).

  1. Fig 5 Images illegible and practically useless, it is not known how the "relative annexin binding" shown in graphs B and D was counted, and it is not known how long the cells were incubated with the tested compounds

  • To address this concern, we divided figure 5 into two sets to give a good resolution. As we stated, the images were taken using a confocal microscope 60x and analyzed using Image J software, and we calculated the intensity by subtracting the background from total fluorescence to correct for the background then divided the fluorescence intensity for every group by the control to give the relative (fold change) intensity compared to the control. The cells were treated for 48 h. (Page 13, page 14, line 440).

  1. Fig 6 The graphs in the flow cytometer are small and the resolution is low. The results do not correspond at all with the photos taken under the fluorescence microscope and the cell viability tests. In the graphs of Figure 6 20uM has minimal effect on the test cells, while in Figures 1 and 5 the changes at 20uM are very significant.

  • In response to the reviewer's comment, we have improved the size and figure resolution. The cell percentage ratio is approximately two times the control matching with the photos taken under the fluorescence microscope and the cell viability results (Page 14, line 460).

  1. Fig 7 Again, very bad resolution, it is not known what is shown in these photos, whether they are whole cells or only parts of their structure. How does the cell viability value ​​in plots C and D correspond to the cell viability values ​​in figure 1? How long were cells with test compounds incubated?

  • To address this comment, we have improved the size and figure resolution. The cells in Fig 7 represent dead cells by anchorage-dependent cell death, also known as anoikis. In this assay, we have used two stains; one stained viable cells (Calcien-AM) and the other stained dead cells (EthD-1) and we compared the number of live cells (stained green) to the dead cells (stained red). However, we couldn’t count the cells in the figures, therefore, we used modified MTT to mimic the anoikis by coating the plate with poly-HEMA to prevent the cells from adhering to the plate (page 5, lines 223 to 240, page 15, line 487). The cells were treated with test compounds for 48 h.

  1. Fig 9 The histograms are unreadable, and the graphs do not show the results of the statistical analysis, although the legend says that it was performed.

  • In response to the reviewer's comment, we have improved the size and figure resolution, and we have added the statistical analysis to the graph (page 17, line 520).

  1. Fig 10 Pictures are illegible and virtually useless, it is not known how the "relative ROS level" shown in graphs B and D were counted, and it is not known how long the cells were incubated with the test compounds.

  • To address this comment, we have improved the size and figure resolution. As stated, the images were analyzed using Image J software, and we have calculated the intensity by subtracting the background from total fluorescence to correct for the background then divided the fluorescence intensity for every group by the control to give the relative (fold change) intensity compared to the control. The cells were incubated with the drug for 6 h (Page 19, line 555).

  1. Fig 11 Histograms are unreadable, C and D plots have unequal scales on the Y-axis, no SD, no statistical analysis performed.

  • In response to the reviewer's comment, we have increased the size and figure resolution. In addition, we have updated the scales for C and D plots. As this experiment (Fig 11) was done during revision of the manuscript to confirm the results that we obtained for ROS immunofluorescence analysis (Fig 10) and due to time and funding constraints we couldn’t perform enough replicates to generate SD or perform statistical analysis (Page 20).

  1. Fig 12 Some of the spots are overexposed e.g., 3a. Therefore, their quantitative analysis may be unreliable, no statistical analysis in the charts and in the description of the statistical analysis in the legend.

  • To address this comment, we have improved the size and figure resolution. The array was imaged using x-ray film processor. The duration of exposure is the same for each and every spot. We have updated the statistics to the graph, and the data was analyzed using Two-way ANOVA method (Page 21, lines 590 to 591).

  1. Fig 153 no SD and statistical analysis in the plots, and the scale of the Y axis in plots C, D, G, H is different, it seems that the experiments shown in the plots in Figs 12 and 13 were not performed in a sufficient number of repetitions.

  • In response to the review's comment, we have updated the scale of the Y axis. Experiments shown in Figs 12 and 13 were done to support the observed anticancer effects and to infer the potential anticancer mechanism of PG in taxane-resistant CRPC cells. It is our intention to follow up on the data presented in Figs 12 and 13 to rigorously delineate the anticancer mechanisms in future studies.

Reviewer 3 Report (Previous Reviewer 3)

In this revised manuscript, the authors have sufficiently addressed my points raised in my initial reports. I support its publication in Cancers. 

I only have very minor suggestion for the authors to consider improving the manuscript: the authors should try keeping fonts in each figure consistent, also in some figures, some of the fonts are actually too small to read (eg. Fig 14)

Author Response

In this revised manuscript, the authors have sufficiently addressed my points raised in my initial reports. I support its publication in Cancers. 

  • We would like to thank the reviewer for supporting our study.
  1. I only have very minor suggestion for the authors to consider improving the manuscript: the authors should try keeping fonts in each figure consistent, also in some figures, some of the fonts are actually too small to read (eg. Fig 14)
  • As suggested, we have unified the text within figures. Also, we have increased the image size  of all figures and text within across the manuscript.

Round 2

Reviewer 2 Report (Previous Reviewer 2)

Although the paper has been substantially revised, there are still some problems that emerged after the last revision. I will refer to the next points of the previous review, which in my opinion still need to be refined. My new remarks are marked with tick - ü

  1. Fig 1 The graphs are small, and the legend shows days shifted incomprehensibly to the right. The X-axis may end at 100uM. But the biggest problem is the fold of induction on the Y axis. What value on Y-axis the X-axis = 0? It looks like it's top 1.2. Therefore, why is the "fold of induction" for 0 for the X axis around 1.2?  What's a "fold of induction" equal to 1?

·        In response to the reviewer's comment, we have improved the size and figure resolution, and adjusted the legend, for the fold induction it was compared to DMSO control (Fold change= 1) and PG concentration in the X-axis corresponds to the fold change in Y-axis (Page 8, line 340 to 341).

 ü  If "fold of induction" means that the cells in Fig 1 increase proliferation upon treatment with the test compound then this should be commented on in the discussion and something about the hormetic effect should be mentioned.

  1. Fig 2 Again, small unreadable charts. How were the IC50 values ​​from Graphs A and B calculated? Charts C and D should be better prepared, and first, at all legible, it isn't easy to deduce anything from them. There are no incubation times in the legend, charts C and D should include names and abbreviations of the inhibitors used. 
  • To address this concern, we have improved the size and resolution of figure 2. The IC50 was determined using GraphPad prism version (9) by calculating the fit dose response curve using non-linear regression. Charts C and D were updated accordingly, and the incubation time is now indicated in the figure legend (Page 9, lines 367 to 368, and page10, line 372).

ü  If The IC50 was determined using GraphPad Prism version (9) by calculating the fit dose-response curve using non-linear regression (like the Authors declare in their reply) why not present dose-response curves prepared using non-linear regression instead of simple charts prepared according to "connect the dots" methods. In line 348-349 the authors write: “Our results reveal that PG significantly decreases cell viability with the increase in PG concentration, as shown in Figures 2A and 2B. In both the cell lines, the half maximal inhibitory concentration (IC50) is 20 μM.” 1st: in Fig 2A IC50 for sure is not 20uM, 2nd, the table with IC50 for both cell lines for each incubation time with SD values ​​should be presented.

 Fig 3 The C and D charts have unequal sizes and, what is worse, the scales on the Y axis. Moreover, it is clearly seen that the number of colonies in the DU145TXR control is greater than in the PCTXR control, the same is in the cells treated with PG20uM + NAC2.5mM while in the plots the values ​​say otherwise. PCTXR control - approximately 680 colonies (according to the scale of the chart) and DU145TXR control about 400 colonies (according to the scale of the chart). It is not known how the colonies could be counted in the DU145 TXR as they merge. The situation is similar in PG20uM + NAC2.5mM. Obviously, the data from the photos do not agree with the data from the charts. The plates like the control for DU145 simply should not be used. Especially as a control

·        To address this comment, we have revised the figure 3. The scales were adjusted, and the control DU145 TXR is updated . To clarify the comment on colony count, it was done manually and confirmed with ImageJ software using the count particle function (Page10, line 386).

 ü  5 Figure 5 looks better, however surprisingly Authors declare that 500 cells were seeded per well since they got much more than 500 colonies in control ...

 Fig 5 Images illegible and practically useless, it is not known how the "relative annexin binding" shown in graphs B and D was counted, and it is not known how long the cells were incubated with the tested compounds

·        To address this concern, we divided figure 5 into two sets to give a good resolution. As we stated, the images were taken using a confocal microscope 60x and analyzed using Image J software, and we calculated the intensity by subtracting the background from total fluorescence to correct for the background then divided the fluorescence intensity for every group by the control to give the relative (fold change) intensity compared to the control. The cells were treated for 48 h. (Page 13, page 14, line 440).

 ü  The description  of the calculation of "relative annexin binding" is still lacking in the materials and methods section, Authors should provide all necessary details to repeat this experiment, including the calculation of "relative annexin binding"

  1. Fig 10 Pictures are illegible and virtually useless, it is not known how the "relative ROS level" shown in graphs B and D were counted, and it is not known how long the cells were incubated with the test compounds.

·        To address this comment, we have improved the size and figure resolution. As stated, the images were analyzed using Image J software, and we have calculated the intensity by subtracting the background from total fluorescence to correct for the background then divided the fluorescence intensity for every group by the control to give the relative (fold change) intensity compared to the control. The cells were incubated with the drug for 6 h (Page 19, line 555).

ü  7 The description of the calculation of " relative ROS level " is still lacking in the materials and methods section, Authors should provide all necessary details to repeated this experiment including the calculation of "relative ROS level"

 Fig 11 Histograms are unreadable, C and D plots have unequal scales on the Y-axis, no SD, no statistical analysis performed.

  • In response to the reviewer's comment, we have increased the size and figure resolution. In addition, we have updated the scales for C and D plots. As this experiment (Fig 11) was done during revision of the manuscript to confirm the results that we obtained for ROS immunofluorescence analysis (Fig 10) and due to time and funding constraints we couldn’t perform enough replicates to generate SD or perform statistical analysis (Page 20).

ü  Fig 11 The histograms are acceptable but there is another interesting point; Authors declare: "Approximately 10,000 cells were seeded with complete RMPI media in a 6-well chamber and incubated at 37 ËšC with 5% CO2 for 48 h. At 70–80%, confluence cells were treated with 20 μM DCFH2-DA and incubated for 45 min. After incubation, cells were treated with different concentrations of PG for 4h" My question is how it is possible that 10 000 cells may reach 70–80%, confluence in well from 6 wells plate in 48h? What is their doubling time?

 Fig 12 Some of the spots are overexposed e.g., 3a. Therefore, their quantitative analysis may be unreliable, no statistical analysis in the charts and in the description of the statistical analysis in the legend.

·        To address this comment, we have improved the size and figure resolution. The array was imaged using x-ray film processor. The duration of exposure is the same for each and every spot. We have updated the statistics to the graph, and the data was analyzed using Two-way ANOVA method (Page 21, lines 590 to 591).

 ü  I am afraid the authors did not understand my remark, I did not mean the resolution but the overexposition. The Authors may read about this problem here https://lukemiller.org/index.php/2013/08/the-dangers-of-overexposing-western-blots/  I'm afraid the gels are overexposed, at least in the most intensive spots. Moreover, t is very interesting how two-way ANOVA has been applied to the statistical analysis. There is no information in the legend about what the asterisk means in the cart, but I guess, this is the significant difference between the control and treated groups. So as I guess the  Authors have two results for each parameter e.g. for Bax or Bcl or each other pair: for the control group they have: mean, SD and n repetitions and for the treated group mean SD and n repetitions.  How for such data two-way ANOWA may be applied? At least my GPPrism gives an error message when I want to perform such a comparison. As I know for ANOWA more than 2 groups are necceseary. Btw what is "n" in your calculations?

  1. Fig 153 no SD and statistical analysis in the plots, and the scale of the Y axis in plots C, D, G, H is different, it seems that the experiments shown in the plots in Figs 12 and 13 were not performed in a sufficient number of repetitions.

·        In response to the review's comment, we have updated the scale of the Y axis. Experiments shown in Figs 12 and 13 were done to support the observed anticancer effects and to infer the potential anticancer mechanism of PG in taxane-resistant CRPC cells. It is our intention to follow up on the data presented in Figs 12 and 13 to rigorously delineate the anticancer mechanisms in future studies.

 I am afraid the Authors did not understand my remark; I asked why there is no statistical analysis and standard deviation in figure 13?

This manuscript is a resubmission of an earlier submission. The following is a list of the peer review reports and author responses from that submission.

Round 1

Reviewer 1 Report

Reshmii V et al reported that Polygodial, a derivative from Tasmanian pepper berry exhibits anti-cancer effect in taxane resistant CRPC cell lines. The functional study nicely showed cell inhibition, colony formation and cell cycle arrest by PG treatment. And the mechanisms behind this finding may due to PG induce cell apoptosis by down-regulating anti-apoptotic genes. I hope the authors could convince me with some in vivo experiments at least showing the efficacy of PG compound and low toxicity in animal study. Still there are some questions arising from in vitro experiments.

1.       Fig.1 shows the toxicity assay of PG treatment in MPCC and 3T3-J2 cells, showing cell viability has no significant changes in two healthy cell lines. MPCC cells best predict the toxicity of liver injury. Have you tried any renal tubule cells for toxicity assay of renal injury? In addition, the quality of finger should be improved. Two figures are not in same size and Day 13 labels are different.

2.       In fig.2. Adding NAC, CI, 3MA and NEC inhibitors could block the effect of PG and increase the cell viability. Can author perform any qPCR or western blots experiments to show NAC, CI,3MA and NEC has any changes after adding PG? Or any rescue experiments to confirm NAC blocks the effect of PG?

3.       In Fig 5 and 6, author use flow cytometry with staining with apoptic maker to confirm this cell apoptosis. It would be better to check if adding NAC, or other inhibiotors could inhibit the apoptosis of cells?

4.       Fig7, qPCR and wester blots could also be the way to double confirm the regulation of ADCD. For fig C and D, cell viability has been showed in Fig2, what is the purpose to show here again?

5.       Fig 11, western blots should was performed to confirm proteome profiler-human apoptosis array results and look for the main dysregulated pathways. Have you tried to do any rescue experiments or any knockdown experiments to find the main target of PG? If PARP-1 is the downstream, author may try PARP-1 inhibitor to see the combine effect?    

Author Response

We thank the reviewers for providing valuable comments that significantly helped improve our manuscript. Our point-by-point responses are provided below.

Reviewer: 1

  1. I hope the authors could convince me with some in vivo experiments at least showing the efficacy of PG compound and low toxicity in animal study.

  • We would like to thank the reviewer for this valuable suggestion. Although we were not able to perform in vivo experiments due to lack of funding, our co-author Prof. Alexander Kornienko has previously published a study in collaboration with De La Chapa et al. in the International Journal of Oncology (PMID: 30320372) that PG could inhibit Cal-27 derived oral squamous carcinoma xenografts in animal model supporting that PG and its analogs have great potential for cancer treatment.

  1. 1 shows the toxicity assay of PG treatment in MPCC and 3T3-J2 cells, showing cell viability has no significant changes in two healthy cell lines. MPCC cells best predict the toxicity of liver injury. Have you tried any renal tubule cells for toxicity assay of renal injury? In addition, the quality of the finger should be improved. Two figures are not of the same size, and Day 13 labels are different.

  • We have not yet tested the toxicity effects of PG on renal tubule cells. However, a study conducted by Fratoni et al. showed that Drimys brasiliensis extract, which contains various polygodial derivatives, did not show hepatic or renal damage in Wistar rats (Fratoni et al., Biomedicine & Pharmacotherapy, PMID: 29864935). Regarding Fig.1 concern, we have improved the quality of the figure as well as corrected the size of the figures and day 13 labels (Page 7, lines325 to 326).

  1. In Fig.2. Adding NAC, CI, 3MA, and NEC inhibitors could block the effect of PG and increase cell viability. Can the author perform any qPCR or western blots experiments to show NAC, CI, 3MA, and NEC has any changes after adding PG? Or any rescue experiments to confirm NAC blocks the effect of PG?

  • As suggested by the reviewer, we conducted a rescue experiment confirming that NAC blocks the effect of PG in colony formation assay, wound healing assay, and ROS production experiments. Furthermore, NAC reversed the effect of PG on anoikis marker PTEN (qPCR- page 15, lines 488-489) and apoptosis marker cleaved PARP1 (Western blot- page 20).

  1. In Fig 5 and 6, the author uses flow cytometry with staining with the apoptotic maker to confirm this cell apoptosis. It would be better to check if adding NAC or other inhibitors could inhibit the apoptosis of cells.
  • Due to time and funding limitations, we could not perform this suggested experiment. However, we intend to thoroughly delineate the various cell death mechanisms of PG in future studies with various inhibitors such as NAC, CI, 3MA, and NEC.

  1. Fig7, qPCR and western blots could also be the way to double confirm the regulation of ADCD. For fig C and D, cell viability has been shown in Fig2. What is the purpose to show here again?

  • As the reviewer suggested, we conducted a qPCR experiment to confirm ADCD, as presented in revised figure 8 (Page 15). Regarding Figures 7 C and D, the MTT was repeated with a different condition as we used poly-HEMA coated plates to mimic the detached condition of cells to study anoikis (Page 5, lines 221 to 237).

  1. Fig 11, western blots should be performed to confirm proteome profiler-human apoptosis array results and look for the main dysregulated pathways. Have you tried to do any rescue experiments or knockdown experiments to find the main target of PG? If PARP-1 is downstream, an author may try PARP-1 inhibitor to see the combined effect.

  • As the reviewer suggested, we conducted a rescue experiment confirming that NAC blocks the effect of PG in colony formation assay, wound healing assay, and ROS production experiments. As the focus of the study is about PG-induced ROS and apoptosis, and to confirm this, we have analyzed PARP1 cleavage as an indicator of apoptosis by PG in the presence and absence of ROS inhibitor (NAC) (Pages 19 and 20, lines 595 to 612). Due to limitations, as indicated before, we will follow up on the proteome profiler-human apoptosis array results to assess the main dysregulated pathways as well as the suggested PARP-1 inhibitor experiment in our future studies.

Reviewer 2 Report

First, the work is not acceptable graphically. Charts and photos have too low resolution and are too small, moreover, they are strangely flattened vertically or horizontally, and descriptions are illegible. Although the article comes from the USA, it requires a linguistic correction.

Line 97 PCa cell lines (PC3-TXR and DU145-TXR) were obtained from ATCC, I’m sorry I could not find these lines in ATCC website, could the Authors provide  ATCC number of these cells?

In the methodology many unusual practices may be found:

Line 97 Further, dilution was made using plain RPMI and stored at -20 ËšC for further use. Could the Authors provide more details on how these dilutions were made and how were added to the incubation mixture?

Line 112 “PCa cells were seeded in 96 well 112 plates at a cell density of 3 x 103/well in 100
μl of complete RPMI (10% FBS, 1% antimycotic, 113 and 0.6% antibiotic). The cells were maintained at a temperature of 37 ËšC with 5% CO2 114 humidity until they reached 50% confluency”. Does it mean that the Authors seeded 3000 cells / well and then evaluated by eye their growth and started experiment when 50% of confluence has been reached?

Line 116 “10
μl of MTT reagent was added 116 to each well after 24 h, 48 h, and 72 h of treatment and incubated for 3 h at 37 °C.” does in mean that MTT has been added to cells without washing with PBS?

After 24 h, the cells were treated with different  concentrations of PG (5
μM, 10 μM, 20 μM, and 50 μM) and incubated for 1- 3 weeks………….. and when the difference in the  colony formation was observed, the media was removed, and the cells were fixed with  acetic acid and methanol in a ratio of 1:7 and incubated at room temperature for 5 minutes.” Does it mean that the Authors compare the samples which are incubated 1 week and 3 weeks?

Line 198 In addition, cell viability was assessed by adding 10% MTT to each well and incubating overnight.

Line 277 why Statistical analysis was done using the Students t-test? In some cases it makes sense in some ANOVA should be used

Fig 1 the charts A and B should be equal, the legend is hard to understand, descriptions of the chart should be unified.

Figure 2 The cytotoxic activity should be presented using GP Prism ,  (the authors declare access this software in line 279 btw number of version should be provided) by nonlinear regression (curve fit) and IC50 values shall be calculated by this way, the IC50 read from the cart provided by the authors may have nothing common with reality. Panels C and D should be also bigger and prepared in the form more friendly for the reader. The results should be described in “Results” section not in legend in the figure

Fig 3 Again low resolution, especially pictures of DU145THR are hard to evaluate, the pictures do not look like taken with 40x magnification like it is declared by  the Authors. These pictures seems to be duplicated in Figure 4.

Figure 4 pictures from scratch assay are lacking, there are only charts, prepared in Excel (Authors should definitively use GP Prism not Excel) with small fonts very hard to read

Figure 5 Again small pictures with low resolution, o not look like from confocal (like Authors declare) charts in prepared in Excel showing “relative annexin V FITC binding” How it was calculated the Authors did not mentioned.

Fig 6 Flow cytometer charts, to small to read and evaluate absolutely useless

Fig 7 In the legend the Authors wrote “We observed that with an increase in the concentration of PG, the relative intensity 450 of calcein-AM (green fluorescence) was decreased, and the relative intensity of EthD-1 (red fluorescence) was increased” Probably the Authors could observe this, for the reader, it is not visible, the picture is useless  

Fig 6 Flow cytometer charts to small to read and evaluate, absolutely useless

Fig 9 Again, the pictures are low resolution; what is the relative ROS level? How was it calculated?

Fig 10 could the Authors explain how the “spotting intensity was evaluated using Image J?

Fig 11 Western Blot; The Authors declare, “Band intensities were analyzed using ImageJ software” could the Authors explain how they did it and what is “relative expression” from Y-axis?   

Author Response

Reviewer: 2

  1. First, the work is not acceptable graphically. Charts and photos have too low resolution and are too small. Moreover, they are strangely flattened vertically or horizontally, and descriptions are illegible. Although the article comes from the USA, it requires a linguistic correction.

  • To respond to the reviewer's comment, we improved the figure quality and corrected the linguistic mistakes across the manuscript.

  1. Line 97 PCa cell lines (PC3-TXR and DU145-TXR) were obtained from ATCC, I’m sorry I could not find these lines on the ATCC website. Could the Authors provide the ATCC number of these cells?

  • We are extremely sorry for overlooking the source of the cell lines. These cell lines were donated to us by Dr. Evan Keller (University of Michigan). We have added this information to the revised manuscript (Page 3, lines 101 to 104).

  1. Line 97 Further, dilution was made using plain RPMI and stored at -20 ËšC for further use. Could the Authors provide more details on how these dilutions were made and how they were added to the incubation mixture?

  • We have prepared the intermediate stock from the main stock of 10mM using complete RPMI media to attain the required concentrations of our drug.

  1. Line 112 “PCa cells were seeded in 96 well plates at a cell density of 3 x 103/well in 100 μl of complete RPMI (10% FBS, 1% antimycotic, and 0.6% antibiotic). The cells were maintained at a temperature of 37 ËšC with 5% CO2 humidity until they reached 50% confluency”. Does it mean that the Authors seeded 3000 cells/well and then evaluated by eye their growth and started an experiment when 50% of confluence had been reached?

  • Yes, we have plated 3000 cells/well, and after visually checking for 50% confluency under bright field light microscope, the cells were treated with respective drugs/inhibitors in this experiment.

  1. Line 116 “10 μl of MTT reagent was added 116 to each well after 24 h, 48 h, and 72 h of treatment and incubated for 3 h at 37 °C.” does in mean that MTT has been added to cells without washing with PBS?

  • As we have observed in our experiments, PG treatment makes the cells loosely adherent to the plate, especially in high concentrations. Therefore, we avoid washing them with PBS to prevent the loss of cells.

  1. After 24 h, the cells were treated with different concentrations of PG (5 μM, 10 μM, 20 μM, and 50 μM) and incubated for 1- 3 weeks…………. and when the difference in the colony formation was observed, the media was removed, and the cells were fixed with acetic acid and methanol in a ratio of 1:7 and incubated at room temperature for 5 minutes.” Does it mean that the Authors compare the samples which are incubated 1 week and 3 weeks?

  • The samples were checked regularly after the first week, and the experiment ended when we observed the control colonies started to merge.

  1. Line 198 In addition, cell viability was assessed by adding 10% MTT to each well and incubating overnight.

  • Our second MTT was conducted with a different condition as we used anchorage resistant plate (Poly-Hema coated wells) to investigate anoikis as per Chinnapaka et al. Free Radical Biology and Medicine, PMID: 31446057]; an overnight incubation is recommended (Page 5, lines 221 to 226).

  1. Line 277 why Statistical analysis was done using the students t-test? In some cases, it makes sense in some ANOVA should be used

  • As the reviewer suggested, we re-analyzed our data using One-way ANOVA and Two-way ANOVA tests followed by Tukey’s multiple comparison test to determine the significance between treatment and control (Page 7, lines 309 to 314).

  1. Fig 1 the charts A and B should be equal, and the legend is hard to understand, descriptions of the chart should be unified.

  • To respond to the reviewer's comment, we have improved the figure quality, legend, and description (Page 7, lines 325-331)

  1. Figure 2 The cytotoxic activity should be presented using GP Prism, (the authors declare access this software in line 279 btw number of versions should be provided) by nonlinear regression (curve fit) and IC50 values shall be calculated by this way, the IC50 read from the cart provided by the authors may have nothing common with reality. Panels C and D should be also bigger and prepared in the form more friendly for the reader. The results should be described in “Results” section not in legend in the figure

  • As suggested by the reviewer, we have added the GP prism version (Page 7, line 313). In addition, we reanalyzed our data using curve fit, panels C and D were also modified, and the figure legend was updated accordingly (Page 9, lines 352 to 361).

  1. Fig 3 Again, low resolution, especially pictures of DU145-TXR are hard to evaluate, the pictures do not look like taken with 40x magnification like it is declared by the Authors. These pictures seem to be duplicated in Figure 4.

  • In response to the reviewer's comment, we have updated figure 3 with the whole image of the plate and updated figure 4 as well (Pages10 and 11).

  1. Figure 4 pictures from scratch assay are lacking. There are only charts prepared in Excel (Authors should definitively use GP Prism, not Excel) with small fonts that are very hard to read

  • In response to the reviewer's comment, we have updated figure 4 (Page 11, lines 395 to 396).

  1. Figure 5 Again, small pictures with low-resolution o does not look like from confocal (like Authors declare) charts in prepared in Excel showing “relative annexin V FITC binding” How it was calculated the Authors did not mention.

  • To respond to the reviewer's comment, we divided figure 5 into two sets to give a good resolution. As we stated, the images were taken using a confocal microscope 60x and analyzed using Image J software, and the obtained fluorescence intensity for every group was divided by the control to give the relative (fold change) intensity compared to the control (Page 12, lines 422 to 431).

  1. Fig 6 Flow cytometer charts, too small to read and evaluate absolutely useless

  • To respond to the reviewer's comment, we have increased the size of the flow cytometer charts (Page 13, lines 445 to 446).

  1. Fig 7 In the legend, the Authors, wrote, “We observed that with an increase in the concentration of PG, the relative intensity of calcein-AM (green fluorescence) was decreased, and the relative intensity of EthD-1 (red fluorescence) was increased” Probably the Authors could observe this, for the reader, it is not visible, the picture is useless
  • To address this concern, we have clarified this point in the manuscript (Pages 14 and 15, lines 464 to 481). As green fluorescence represents viable cells, while red fluorescence represents dead cells.

  1. Fig 6 Flow cytometer charts too small to read and evaluate, absolutely useless

  • To address this concern, we have increased the size of the flow cytometer charts for better resolution (Page 13).

  1. Fig 9 Again, the pictures are low resolution; what is the relative ROS level? How was it calculated?

  • To respond to the reviewer's comment, we updated the images with new ones (now Fig 10) taken at 40x and analyzed them using Image J software. The mean of the result fluorescence intensity for every group was divided by the control to give the relative (fold change) intensity compared to the control (Page 17, lines 536 to 546).

  1. Fig 10 could the Authors explain how the “spotting intensity was evaluated using Image J?

  • Fig 10 is now Fig 12 in the revised manuscript. The spot intensity was analyzed using Image J software using a specific Java macro developed by Gilles Carpentier, 2008. The macro is accessible at (https://imagej.nih.gov/ij/macros/toolsets/Dot%20Blot%20Analyzer.txt), and we cited the tool and provided detailed information (Pages 6 and 7, lines 290 to 293).

  1. Fig 11 Western Blot; The Authors declare, “Band intensities were analyzed using ImageJ software” could the Authors explain how they did it and what is “relative expression” from Y-axis?

  • Western Blot analysis was done using ImageJ software, and we followed the same procedure provided by the image J website (https://imagej.nih.gov/ij/docs/menus/analyze.html#gels). The relative intensity was calculated by dividing the relative intensity by the one we get with β-actin normalization. In addition, to address this concern, every data point was divided by the control to give the final fold change in the expression (Page 20).

Reviewer 3 Report

In this submitted research article titled “Polygodial, a sesquiterpene of Tasmanian pepper berry, activates apoptotic signaling in castration-resistant prostate cancer cell lines via inducing oxidative stress”, Venkatasan et al presented here polygodial (PG), a natural product derived from Tasmanian pepper berry, robustly inhibits the cell viability, colony formation, and migration of taxane resistant CRPC (PC3-TXR and DU145-TXR) cell lines while exhibits no toxicity to primary human hepatocytes and 3T3-J2 fibroblast co-cultures. The authors further revealed PG displays the ability to downregulate anti-apoptotic markers like Bcl-2, Bcl-x and activates cytochrome c and caspase 3. In addition, the authors showed that PG can induce ROS in the tested PC3-TXR and DU145-TXR cells. 

This research article provided seemingly intriguing results to support that PG displays promising therapeutic potential in taxane-resistant CRPC cell lines. I am adding below a list of questions and suggestions for the authors to consider and I believe these will help to improve this research article.

Major comments:

1: taxol should be used as a control in the PC3-TXR and DU145-TXR cells in several assays, such as the viability assay, colony formation assay, Annexin V/PI assay…

2: For the colony formation assay, full instead of selected partial pictures for the whole wells should be shown. This is because PC3-TXR and DU145-TXR cells are prone to grow along the side of wells, therefore it will be hard to tell cell wipeout only based on a cropped picture.

3: the authors need to upload the right figure 4 (migration assay)

4: according to figures 1 C and D, it seems the cytotoxicity induced by PG can be fully rescued by autophagy inhibitor, necroptosis inhibitor, what is the authors` explanation?

5: Figure 5 is not clear enough.

6: the authors should provide western blots result for cleaved caspase 3 and cleaved PARP, which are more specific apoptosis biomarkers than the pro-forms.

7: the authors claimed cytotoxicity induced by PG mainly resulted from induction of ROS, an important experimental result that whether commonly-used reductants can rescue PG-induced cell death should be provided.

Author Response

 Reviewer: 3

  1. Taxol should be used as a control in the PC3-TXR and DU145-TXR cells in several assays, such as the viability assay, colony formation assay, Annexin V/PI assay.

  • We appreciate the reviewer's suggestion. It has been shown that these cell lines are resistant to taxanes, as per Takeda et al. [ Prostate, 2007, PMID: 17440963]. These taxol-resistant cell lines were a kind gift from Dr. Evan Keller’s lab (University of Michigan). We have also reconfirmed the taxol resistance of these cell lines by conducting a cell viability assay with various doses of taxol (data not shown). Since these cell lines display taxol resistance, we have not used it as a control in our experiments.

  1. For the colony formation assay, full instead of selected partial pictures for the whole wells should be shown. This is because PC3-TXR and DU145-TXR cells are prone to grow along the side of wells. Therefore, it will be hard to tell cell wipeout only based on a cropped picture.

  • In response to the reviewer's comment, we have updated figure 3 with a full image of the plates (Page 10, lines 371 to 372).

  1. The authors need to upload the right figure 4 (migration assay)

  • In our original submission, scratch assay images were inadvertently omitted during file uploading. We are sorry for this error. We have updated figure 4 with relevant pictures (Page 11, lines 395 to 396).

  1. According to figures 1 C and D, it seems the cytotoxicity induced by PG can be fully rescued by autophagy inhibitor, necroptosis inhibitor, what is the authors` explanation?

  • Our data presented in figures 1 C and D suggest that PG, in addition to apoptosis, could also induce autophagy and necroptosis, which needs further evaluation.

  1. Figure 5 is not clear enough.

  • To respond to the reviewer's comment, we divided figure 5 into two sets to give a good resolution (Page 12).

  1. The authors should provide western blots result for cleaved caspase 3 and cleaved PARP, which are more specific apoptosis biomarkers than the pro-forms.

  • The authors would like to thank the reviewer for this suggestion. Given the time and funding constraints of the suggested apoptotic markers, we have provided the results of cleaved PARP (Page 20, lines 601 to 612). In future studies, we would like to follow up on the detailed anticancer mechanisms of PG.

  1. The authors claimed cytotoxicity induced by PG mainly resulted from induction of ROS, an important experimental result that whether commonly used reductants can rescue PG-induced cell death should be provided.

  • As the reviewer suggested, we conducted a rescue experiment confirming that NAC blocks the effect of PG in colony formation assay, wound healing assay, and ROS production experi Due to limitations, as indicated before, we could not perform the suggested rescue experiment to show that reductants such as NAC could rescue PG-induced cell death. However, it is our priority to thoroughly delineate the various cell death mechanisms of PG in future studies.

Reviewer 4 Report

1. The manuscript could benefit from editing for grammar, missing words, and subject verb agreement, etc. It is recommended that authors delete irrelevant "general" phrases and sentences, repeated and unneeded words. They should use short sentences. Also, some Introductory sentences are irrelevant or are not needed. For example, in the simple summary, “The emergence of resistance to androgen deprivation therapy (ADT) resulted in …” should be “The emergence of resistance to androgen deprivation therapy (ADT) results in.” Also, in the abstract, “These have poor outcomes and resulted in” should be changed to “These have poor outcomes and result in.”

2. In the simple summary, authors mentioned that “Patients with CRPC treated with taxanes show poor outcomes.” This statement does not relate to the preceding or the following sentence. What is the relation between CRPC and taxanes, and what do taxanes have to do with sesquiterpenes? Kindly modify this section to have a proper flow of ideas. Same applies to the abstract. Maybe, giving examples of taxanes would help give an idea about what this drug is (chemotherapy?).

3. Abstract: “Toxicity investigation shows that PG is not toxic to primary human hepatocytes and 3T3-J2 fibroblast co-cultures, implicating that PG is innocuous to healthy cells.” What about normal prostate cell lines? Did authors assess whether PG is toxic to normal prostatic epithelial cells?

4. Keywords: I suggest avoiding the use of acronyms (ADCD) here as not all readers are familiar with such terms.

5. All abbreviations should be revised and defined at their first use (such as USA in the introduction). Same applies to other abbreviations throughout the manuscript. Also, once an acronym is used, there is no need to write the whole word again (such as “localized prostate cancer” on line 68).

6. Introduction: “1 in 8 men will have prostate cancer during their lifetime.” In scientific writing, it is recommended not to start a sentence with a numeric number but rather write the whole word: One in eight men …

7. Introduction: “with PCa ranked fourth among the top 5 most common cancers, with 7.3% of all new PCa cases coming after breast cancer (11.7%), lung cancer (11.4%), colorectal cancers (10.0 %).” This sentence needs to be rephrased. What is meant by “with 7.3% of all new PCa cases coming after …”? Maybe the authors meant that 7.3% of all new cancer cases are PCa.

8. Introduction: “which mainly originated from spatial,” should be corrected “which mainly originates from spatial.” The use of past and present tenses should be revised in the whole manuscript. When presenting facts, authors need to use the present tense (an example is on line 76 “Predominantly resistance to ADT therapy emerged within 2 to 3 years of treatment” where the present tense should have been used).

9. Introduction: “ADT in combination with LHRH agonist-antagonist along with taxanes” kindly give examples about taxanes.

10. Introduction: “and the gram-positive bacteria, such” remove “the.”

11. This heading needs to be corrected “2.1. Prostate cancer cell line”: lines instead of line.

12. Methods: In the cell lines section, please include the other cell lines used such as hepatocyte/3T3-J2 fibroblast MPCCs and 3T3-J2 fibroblast.

13. Methods: What is the dilution % of the drug in DMSO? i.e. Was it reconstituted in 0.1% dimethyl suldoxide? (which usually does not cause toxicity).

14. Methods: For the cell viability MTT assay, why did authors choose to culture cells for 6 days? Cell viability assay should use different time points at which media is removed and replaced with fresh media along with 10μL/well of the MTT yellow dye on each time point.

15. Methods: A major issue with the wound healing assay is that authors should have used Mitomycin C. Mitomycin C, a cellular proliferation inhibitor, is used prior to the wound formation to ensure wound closure is due to cell migration and not cell proliferation. Those experiments need to be repeated with the use of Mitomycin C.

16. Methods: “The chambers were removed from the slide and after washing with 1X PBS.” This sentence is incomplete.

17. Methods: The whole methods section is very detailed. For example, authors can shorten the Western Blotting section as long as they are citing the protocol from which their experiments were done.

18. Results: Authors have treated versus untreated conditions for all experiments. However, did they include the vehicle (DMSO) condition in their experiments?

19. Figure 1 legend: “does not significantly affect by PG treatment.” This statement needs to be corrected and rephrased: “is not significantly affected by PG treatment.”

20. All the figure legends can be revised as to be more informative of the images presented. For example, adding the statistical test used in the figure legend would be beneficial.

21. In scientific writing, in general, symbols for genes are italicized whereas symbols for proteins are not italicized. The formatting of symbols for RNA and complementary DNA (cDNA) usually follows the same conventions as those for gene symbols. Gene names that are written out in full are not italicized (e.g., insulin-like growth factor 1). Genotype designations should be italicized, whereas phenotype designations should not be italicized.

22. Figure 4: I suggest adding bright field images showing how the drug is inhibiting wound closure and migration of cells.

23. The discussion is very well written.

Author Response

Reviewer: 4

  1. The manuscript could benefit from editing for grammar, missing words, and subject verb agreement, etc. It is recommended that authors delete irrelevant "general" phrases and sentences, repeated and unneeded words. They should use short sentences. Also, some Introductory sentences are irrelevant or are not needed. For example, in the simple summary, “The emergence of resistance to androgen deprivation therapy (ADT) resulted in …” should be “The emergence of resistance to androgen deprivation therapy (ADT) results in.” Also, in the abstract, “These have poor outcomes and resulted in” should be changed to “These have poor outcomes and result in.”
  • As the reviewer suggested, we have corrected the grammatical errors across the manuscript.

  1. In the simple summary, authors mentioned that “Patients with CRPC treated with taxanes show poor outcomes.” This statement does not relate to the preceding or the following sentence. What is the relation between CRPC and taxanes, and what do taxanes have to do with sesquiterpenes? Kindly modify this section to have a proper flow of ideas. Same applies to the abstract. Maybe, giving examples of taxanes would help give an idea about what this drug is (chemotherapy?).

  • As the reviewer suggested, we have updated the flow of this section in the simple summary and abstract. In addition, we have indicated that taxanes are diterpene-type compounds and have given some examples (Page 1, lines 17 to 36).

  1. Abstract: “Toxicity investigation shows that PG is not toxic to primary human hepatocytes and 3T3-J2 fibroblast co-cultures, implicating that PG is innocuous to healthy cells.” What about normal prostate cell lines? Did authors assess whether PG is toxic to normal prostatic epithelial cells?

  • To respond to the reviewer's comment, we have assessed the effects of PG on non-malignant BPH-1 cell lines. Results showed that PG is relatively non-toxic to normal prostatic epithelial cells (Pages 8 and 9, lines 345 to 361).

  1. Keywords: I suggest avoiding the use of acronyms (ADCD) here as not all readers are familiar with such terms.

  • The authors would like to thank the reviewer for this suggestion. To respond to the reviewer's comment, we have used the familiar term “Anoikis” instated throughout the manuscript.

  1. All abbreviations should be revised and defined at their first use (such as the USA in the introduction). Same applies to other abbreviations throughout the manuscript. Also, once an acronym is used, there is no need to write the whole word again (such as “localized prostate cancer” on line 68).

  • As the reviewer suggested, we have updated the abbreviations across the manuscript.

  1. Introduction: “1 in 8 men will have prostate cancer during their lifetime.” In scientific writing, it is recommended not to start a sentence with a numeric number but rather write the whole word: One in eight men …

  • To respond to the reviewer's comment, we have corrected that sentence (Page 2, line 51).

  1. Introduction:“with PCa ranked fourth among the top 5 most common cancers, with 7.3% of all new PCa cases coming after breast cancer (11.7%), lung cancer (11.4%), colorectal cancers (10.0 %).” This sentence needs to be rephrased. What is meant by “with 7.3% of all new PCa cases coming after …”? Maybe the authors meant that 7.3% of all new cancer cases are PCa.

  • As the reviewer suggested, we have updated the flow of this paragraph, removing the confusing information (Page 2, lines 50-54).

  1. Introduction:“which mainly originated from spatial,” should be corrected “which mainly originates from spatial.” The use of past and present tenses should be revised in the whole manuscript. When presenting facts, authors need to use the present tense (an example is on line 76 “Predominantly resistance to ADT therapy emerged within 2 to 3 years of treatment” where the present tense should have been used).

  • The authors would like to thank the reviewer for these suggestions. To respond to the reviewer's comment, we have corrected grammar across the manuscript.

  1. Introduction:“ADT in combination with LHRH agonist-antagonist along with taxanes” kindly give examples about taxanes.
  • As the reviewer suggested, we have given examples for taxanes (Page 2, lines 73-75).

  1. Introduction:“and the gram-positive bacteria, such” remove “the.”

  • To respond to the reviewer's comment, we have removed “the” From the gram-positive bacteria (Page 2, line 93)

  1. This heading needs to be corrected “1. Prostate cancer cell line”: lines instead of line.

  • As the reviewer suggested, we have changed the heading name to cell lines (Page 3, line 100).

  1. Methods:In the cell lines section, please include the other cell lines used, such as hepatocyte/3T3-J2 fibroblast MPCCs and 3T3-J2 fibroblast.

  • We are sorry for not providing the information on other cell lines used in the study. We have now updated this information to the revised manuscript (Page 3, lines 106 to 117).

  1. Methods: What is the dilution % of the drug in DMSO? i.e., Was it reconstituted in 0.1% dimethyl suldoxide? (Which usually does not cause toxicity).

  • The percentage of DMSO used was < 0.1%.

  1. Methods:For the cell viability MTT assay, why did authors choose to culture cells for 6 days? Cell viability assay should use different time points at which media is removed and replaced with fresh media along with 10μL/well of the MTT yellow dye on each time point.

  • As indicated in the method section, Hepatocyte/3T3-J2 fibroblast MPCCs take a prolonged time to fill the micropatterned co-culture system. Therefore, we started to treat these cells at day 7, and to investigate the long-term effect of PG in those cells, we added the drug every time we changed the media and measured the viability at days 9,11, and 13 as per Berger et al. (Hepatology, 2015, PMID: 25421237).

  1. Methods:A major issue with the wound healing assay is that authors should have used Mitomycin C. Mitomycin C, a cellular proliferation inhibitor, is used prior to the wound formation to ensure wound closure is due to cell migration and not cell proliferation. Those experiments need to be repeated with the use of Mitomycin C.

  • As the reviewer suggested, we have pretreated the cells with Mitomycin C in wound healing assay using 15µM concentration and incubated for 1h, as we stated in the methods section (Page 4, lines179 to 182).

  1. Methods:“The chambers were removed from the slide and after washing with 1X PBS.” This sentence is incomplete.
  • To respond to the reviewer's comment, we have completed the sentence (Page 6, lines 261 to 264)

  1. Methods: The whole methods section is very detailed. For example, authors can shorten the Western Blotting section as long as they cite the protocol from which their experiments were done.

  • As the reviewer suggested, we have shortened the method section and have provided appropriate references (Page 7, lines 294 to 308).
  1. Results: Authors have treated versus untreated conditions for all experiments. However, did they include the vehicle (DMSO) condition in their experiments?
  • The percentage of DMSO used was < 0.1%, which is not toxic to the cells, and therefore vehicle control was not used separately in our experiments.

  1. Figure 1 legend: “does not significantly affect by PG treatment.” This statement needs to be corrected and rephrased: “it is not significantly affected by PG treatment.”

  • As the reviewer suggested, we have corrected the statement (Page 8, lines 330 to 331).

  1. All the figure legends can be revised to be more informative of the images presented. For example, adding the statistical test used in the figure legend would be beneficial.

  • We appreciate the reviewer's suggestion. However, we used one-way ANOVA followed by Tukey’s test for comparison between groups, as we stated in the method section (page 7; lines 310-314). Therefore, we think that adding this information to every legend will be a repetition.

  1. In scientific writing, in general, symbols for genes are italicized, whereas symbols for proteins are not italicized. The formatting of symbols for RNA and complementary DNA (cDNA) usually follows the same conventions as those for gene symbols. Gene names that are written out in full are not italicized (e.g., insulin-like growth factor 1). Genotype designations should be italicized, whereas phenotype designations should not be italicized.

  • As the reviewer suggested, we have corrected this information across the manuscript.

  1. Figure 4:I suggest adding bright field images showing how the drug is inhibiting wound closure and migration of cells.

  • To respond to the reviewer's comment, we have updated figure 4 (Page 11, lines 395 to 396).

  1. The discussion is very well written.

  • We would like to thank the reviewer for acknowledging our effort.